



# Influence of light absorbing particles on snow spectral irradiance profiles

Francois Tuzet[1,2], Marie Dumont[1], Laurent Arnaud[2], Didier Voisin[2], Maxim Lamare[1], Fanny Larue[2],
Jesus Revuelto[1], and Ghislain Picard[2]

[1]Univ. Grenoble Alpes, Université de Toulouse, Météo France, CNRS, CNRM, Centre d'Étude de la Neige, Grenoble, France
[2]UGA,CNRS, Institut des Geosciences de l'Environnement (IGE) UMR 5001, Grenoble, France

**Correspondence:** Francois Tuzet (francois.tuzet@meteo.fr)

**Abstract.**

Light Absorbing Particles (LAP) such as black carbon or mineral dust are some of the main drivers of snow radiative transfer. Small amounts of LAP significantly increase snowpack absorption in the visible wavelengths where ice absorption is particularly weak, impacting the surface energy budget of snow-covered areas. However, linking measurements of LAP concentration in snow to their actual radiative impact is a challenging issue which is not fully resolved. In the present paper, we point out a new method based on Spectral Irradiance Profile (SIP) measurements which makes it possible to identify the radiative impact of LAP on visible light extinction in homogeneous layers of the snowpack. From this impact on light extinction it is possible to infer LAP concentrations present in each layer using radiative transfer theory. This study relies on a unique dataset composed of 26 spectral irradiance profile measurements in the wavelength range 350-950 nm with concomitant profile measurements of snow physical properties and LAP concentrations, collected in the Alps over two snow seasons in winter and spring conditions. For 55 homogeneous snow layers identified in our dataset, the concentrations retrieved from SIP measurements are compared to chemical measurements of LAP concentrations. A good correlation is observed for measured concentrations higher than 5 ng $g^{-1}$ ($r^2 = 0.74$ ) despite a clear positive bias. The potential causes of this bias are discussed, underlining a strong dependence of our method to LAP optical properties and to the relationship between snow microstructure and snow optical properties used in the theory. Additional uncertainties such as artefacts in the measurement technique for SIP and chemical contents along with LAP absorption efficiency, may explain part of this bias. In addition, spectral information on LAP absorption can be retrieved from SIP measurements. We show that for layers containing a unique absorber, this absorber can be identified in some cases (e.g: mineral dust vs black carbon). We also observe an enhancement of light absorption between 350 and 650 nm in presence of liquid water in the snowpack which is discussed but not fully elucidated. A single SIP acquisition lasts approximately one minute and is hence much faster than collecting a profile of chemical measurements. With the recent advances in modelling LAP-snow interactions, our method could become an attractive alternative to estimate vertical profiles of LAP concentrations in snow.



# 1 Introduction

Snow is a highly reflective medium in the visible and near infrared (referred to as NIR) wavelengths where most of the solar energy is available (Warren, 1982). The amount of solar energy absorbed by snow-covered areas is hence small compared to other surfaces such as bare soil, vegetation or oceans, making snow a singular component of our climate system (Armstrong and Brun, 2008). Snow optical properties depend on its physico-chemical characteristics whose evolution is driven by atmospheric conditions (Colbeck, 1982; Aoki et al., 2006). This dependence involves snow in strong optical feedback loops that are of crucial importance for the snowpack evolution and are still poorly understood (Hall, 2004; Box et al., 2012). Light Absorbing Particles (LAP) in snow, such as mineral dust (referred to as dust in the following; Di Mauro et al., 2015), Black Carbon (BC; Painter et al., 2013) or algae (Cook et al., 2017), trigger and amplify these snow albedo feedbacks, impacting significantly the cryosphere and its evolution under a changing climate (Skiles et al., 2018).

Linking snow albedo to snow physical properties and LAP concentrations has been an active field of research over the last decades (e.g., Wiscombe and Warren, 1980; Hadley and Kirchstetter, 2012; Skiles, 2014; Adolph et al., 2017). Nowadays the underlying theory is well-known (Warren, 1982) and many radiative transfer models are able to numerically compute snow optical properties for given physical properties and LAP concentrations (e.g., Flanner and Zender, 2006; Aoki et al., 2011; Tuzet et al., 2017). However, from a practical point of view, modelling the impact of LAP on the optical properties of snow still remains challenging due to several issues. Firstly, chemical analyses of snow samples to determine concentrations and size distributions of LAP are time consuming and suffer from intrinsic limitations, since most analytical techniques are only sensitive to certain particle sizes. In the case of BC, where direct determinations which are only sensitive to small size particles coexist with filtration based techniques mostly sensitive to larger size particles, Schwarz et al. (2012, 2013) estimates that the resulting uncertainties on total BC concentrations in snow can be as high as 60%. Secondly, the radiative impact of a given concentration of LAP is highly uncertain due to strong variations of the LAP intrinsic optical characteristics driven by their physical (e.g. size distribution, density, aging) and chemical (e.g. coating, hygroscopicity) properties. Coating of LAP by non-absorbing aerosols is, for example, suspected to enhance their absorption efficiency by up to a factor 3 (e.g., Schnaiter et al., 2005; Moffet and Prather, 2009). Caponi et al. (2017) highlighted the high variability of the optical properties of dust particles with respect to their size distribution and their origin, leading to one order of magnitude uncertainty in absorption by dust for a given mass. Thirdly, the interactions between LAP and snow are known to impact LAP absorption efficiency but are still poorly understood. Flanner et al. (2012) highlighted that for a given BC concentration in snow, the absorption can be up to twice as much if particles are inside the ice rather than in the air surrounding the ice, but estimating LAP mixing state is challenging. Moreover, knowledge about the impact of LAP-snow interactions on other particle properties such as size distribution, coating or hygroscopicity is still at an early stage. Dong et al. (2018) recently revealed that more particles are coated by other species in snow and ice than in the atmosphere, but the impact on radiative transfer has not yet been evaluated. All these issues have been reported for years (e.g., Doherty et al., 2010; Flanner et al., 2012; He et al., 2017) and are still unsolved, mostly due to the difficulty to observe LAP in snow with simultaneous measurements of their optical properties.





Determining LAP absorption in snow is a complex experimental problem which can difficultly be addressed with a direct approach such as joint measurements of chemical concentrations and albedo. Indeed, not only do chemical measurements present high uncertainties as mentioned above, but albedo measurements also have uncertainties of their own, hindering the detection of the effect of LAP on albedo at low concentrations (Warren, 2013). Even at higher concentrations, the precise vertical

distribution of the LAP in the uppermost millimeters is crucial for an accurate estimation of albedo. However, sampling snow with such a high vertical resolution in snowpits is rarely achieved. Recent studies based on hyperspectral (e.g., Dal Farra et al., 2018) or TEM–EDX (e.g., Dong et al., 2018) microscopy bring an understanding of the physico-chemical properties of LAP in snow at the particle scale but remain difficultly applicable to a large number of samples. To date, the understanding of LAP absorption efficiency in snow remains strongly uncertain although it is a crucial parameter to accurately model their impact on

the cryosphere.

In this study, we propose an alternative approach, based on Spectral Irradiance Profile (SIP) measurements in snow, from which snow extinction can be retrieved and compared to the expected optical impact of LAP. Even if most of the energy is absorbed in a very thin top layer (few millimeters; Brandt and Warren, 1993; Libois et al., 2014), understanding light penetration is of crucial importance for the thermal regime of the snowpack (Flanner and Zender, 2005; Picard et al., 2012), for photosynthetic activity

of underlying vegetation (Richardson and Salisbury, 1977) and for in-snow photochemistry (Grannas et al., 2007; Domine et al., 2008; France et al., 2012). Light penetration and transmittance measurements in snow started with Liljequist (1956). Section D.3 in Warren (1982) summarises available measurements at that time. They were mostly limited to monochromatic or spectrally integrated radiation. More recently, spectrally resolved irradiance profiles have been measured in the UV and visible for photochemistry purposes (e.g., King et al., 2001; France et al., 2012). In addition to their SIP measurements, France et al.

(2012) had concomitant chemical measurements of carbonaceous species (Voisin et al., 2012). They observed that measured LAP concentrations were too low to explain the absorption of the snowpack in the visible assuming state-of-the-art LAP absorption efficiencies.

A few studies have undertaken comparisons between SIP measurements and radiative transfer theory. Libois et al. (2013, 2014) measured SIP in the visible and NIR to determine the absorption enhancement factor related to the shape of the ice

crystals in snow. Warren et al. (2006) and Picard et al. (2016) refined the absorption spectrum of pure ice by combining SIP measurements and radiative transfer theory relying on the absence of LAP in Antarctic snow. Picard et al. (2016) suggested that BC traces as low as 5 ng g$^{-1}$ have a detectable effect on SIP measurements, meaning that SIP measurements could be an order of magnitude more sensitive to LAP than albedo measurements. It is consistent with the study of Reay et al. (2012) that highlighted that OH and $NO_2$ production in depth is strongly impacted by small changes of LAP concentration in snow.

Accounting for LAP when modelling light penetration in snow is hence of the utmost importance even when concentrations are too low to significantly impact albedo.

This paper investigates the relationship between SIP measurements and chemically-measured LAP concentrations in snow to assess the absorption efficiency of LAP. To this end, 26 SIP measurements acquired in the French Alps are analysed using a radiative transfer model. LAP concentrations and snow physical properties explaining the spectral signature of SIP measurement

are compared to in-situ measurements. The uncertainties affecting the measurements and model parameters are also investigated.





Section 2 presents the measurement dataset consisting of combined measurements of SIP, snow physical properties and chemical measurements profiles of BC, and dust concentrations. Section 3 details the processing applied to the SIP measurements and the method used to compare them with radiative transfer modelling. Finally the results are presented in Section 4 and limitations of the method are discussed in Section 5.

## 2   Data and study site

Data were measured over 33 days during two winter seasons in 2016-2017 and 2017-2018 at the Col du Lautaret site (45°02'28.7"N 6°24'38.0"E) around 2100 m a.s.l. in the French Alps. This unique dataset includes SIP measurements with snow physico-chemical properties from a coincident snowpit. All the field sampling and measurements were performed by a single operator for the two seasons ensuring a stable protocol, detailed in the following section. The dataset spans across a wide range of meteorological, illumination and snow conditions as the measurements were taken both in winter and spring conditions from the onset to the total melt-out of the snowpack.

### 2.1   Spectral Irradiance Profiles (SIP)

Up to three SIP were collected each day on a flat, horizontal and unaltered snow surface using the SOLEXS (SOLar EXtinction in Snow; Libois et al., 2014) instrument, which consists in a fiber optic connected to a spectrometer. A full description and schematic illustrations of the instrument can be found in Libois et al. (2014) and in section 2.1 of Picard et al. (2016).

First, a vertical hole of 10 mm diameter is drilled by inserting a metal rod up to a depth of 50 cm depending on the presence of hard layers in the snowpack. Second, the fiber optic, fitted in a white rod, is slowly inserted in this hole, taking precautions not to enlarge the hole. The depth of the fiber is precisely measured with a magnetic coding ruler with 1 mm resolution. The fiber transmits light to a spectrophotometer operating in the spectral range 300–1100 nm with 3 nm spectral resolution. A spectrum is acquired every 5 mm during descent and ascent ensuring a 5 mm vertical resolution or better. An acquisition takes from 7 ms to 1000 ms depending on the overall irradiance, which is mainly a function of depth. In total a two-way profile is completed in about a minute, a period during which the incoming radiation can vary. A photosensor is placed at the surface to record variations of broadband incident irradiance in order to detect large variations, and allow the correction of small variations.

The SIP for which incident irradiance had varied more than 3% during the measurements were discarded. Spectral data at wavelengths less than 350 nm and more than 950 nm are usually very noisy and are not exploited here, because of the sharp decrease of the irradiance with wavelength in the NIR, associated with the low sensitivity of the spectrometer in that range, as well as the limited incoming radiation in the UV. When the operator begins the acquisition of the SIP, the magnetic ruler measurement is set to 0 in order to acquire depth from the top of the snowpack. For 6 profiles of the whole dataset, a vertical offset of a few millimeters was introduced in the SIP measurement during operation. By visualizing the profiles, we applied an ad hoc correction by taking the first point where irradiance start decreasing as $z = 0$. In total, these 6 profiles were corrected by an offset smaller than 2 cm.





## 2.2 Snowpit data

Vertical profiles of snow physical properties were collected at the exact position where the SIP was acquired. When multiple SIP acquisitions were performed on the same day, the physical properties were collected in between the different SIP, which were never separated by more than 50 cm. In the snowpit, density was measured at a 6 cm vertical resolution using a cylindric

cutter with a volume of 0.5 L. Ice layers were excluded for practical reasons. Following Proksch et al. (2016), who suggest an uncertainty on density measurements between 2 and 5%, we consider a 5% relative uncertainty in our measurements. Specific Surface Area (SSA) vertical profiles were also collected. During the snow season 2016-2017, these profiles were measured with the DUFISSS instrument (Gallet et al., 2009), with one sample every 3 cm, excluding ice layers. During the snow season 2017-2018, SSA profiles were measured with the ASSSAP profiler, which is a lightweight version of POSSSUM (Arnaud et al.,

2011; Libois et al., 2014). Over both seasons, measured SSA values range from 5 to 55 $m^2$ $kg^{-1}$. For this range of values, Gallet et al. (2009) and Arnaud et al. (2011) suggest that the DUFISSS and ASSSAP accuracy is around 10%. It is to note that Arnaud et al. (2011) also realised an inter-calibration of these two instruments and obtained a 6% RMS difference. Concomitant measurements of temperature, wetness, and snow grain type according to Fierz et al. (2009) were also performed in snowpits.

## 2.3 Chemical analyses

The vertical profiles of dust and refractory Black Carbon (rBC) concentration were measured with a 3 cm vertical resolution on the samples taken from the uppermost 20 cm of the snowpack at least. Snow was sampled in triplicates in sterile 50 mL polypropylene centrifugation tubes with extra care to avoid any contamination by the operator. The samples remained frozen until analysis, avoiding freeze/thaw cycles suspected to impact LAP size distribution in snow (e.g., Lim et al., 2014; Schwarz et al., 2013). In the laboratory, BC was analysed immediately after melting as rBC, using a Single Particle Soot Photometer

($SP^2$, Droplet Measurement Technologies). Samples were nebulised and the resulting aerosol was analysed in the laboratory following the procedure described in Wendl et al. (2014). External calibration samples with freshly prepared Aquadag standards were run before each sample series. As the size distribution of the Aquadag samples was close to the size distribution of BC in snow, the nebulisation biases between standards and samples were minimised. Typical analytical repeatability and calibration uncertainties cumulate to ∼5 %, but this does not account for potential nebulisation biases due to dissimilarities between the

size distributions of BC in snow and the standards. The nebuliser used in the analysis causes potential maximum biases up to 20%. The maximum uncertainty of rBC measurements combining in quadrature nebulisation biases, calibration uncertainties and repeatability is estimated to ∼21%.

Dust size distributions and concentrations were measured with a Coulter Counter following Delmonte et al. (2004) . The measured sizes span a range of 0.6 to 21 μm, in 256 logarithmically spaced size bins. Coulter Counter counts and measures

insoluble particles, so we assume here that insoluble particles above 0.6 μm are mainly dust particles, which agrees well with the measured volume weighted average size for our measurements (typically 1.2 to 4 μm). Depending on initial concentrations, samples were diluted by a factor of up to 100 and the blank concentration was subtracted. This correction stayed below 7 % for 3/4 of samples. The Coulter Counter measurement total uncertainty for dust concentration is estimated to ∼10%.





## 3 Methods

### 3.1 LAP concentration

Since several LAP types are present in the snowpack at any time over a season, it is convenient to present results in term of effective optically equivalent BC (eqBC) concentrations $c_{eqBC}$ as in Dumont et al. (2017). For both measured and estimated

LAP concentrations, the eqBC concentration is calculated as:

$$c_{\mathrm{eqBC}} = c_{\mathrm{BC}} + \psi(c_{\mathrm{dust}}), \tag{1}$$

where $c_{\mathrm{BC}}$ is the BC concentration, $c_{\mathrm{dust}}$ is the dust concentration. $\psi$ is a function computing the BC concentration that would have the same integrated radiative impact from 350 to 900 nm as the input dust concentration (Figure 1). The radiative impact is computed considering a spectral incoming irradiance repartition for mid-latitude winter in clear sky conditions computed with

the detailed atmospheric radiative model SBDART (Ricchiazzi et al., 1998). It is to note that the function $\psi$ depends on the radiative transfer model parameters, mainly on the selected values of BC and dust Mass Absorption Efficiency (MAE). In the following, the unit ng g$^{-1}$ eqBC refers to 1 ng g$^{-1}$ of eqBC concentration. Concerning the measurements, the concentration of a layer is computed as the mean of all concentration measurements in this layer, weighted by the measured density associated to this layer. As BC in our snow samples is analysed as rBC, we use the abbreviation eqrBC for measurements. Using eqBC makes

it possible to represent all LAP impacts with a single number, which is clearer but comes with assumptions that must be kept in mind for the interpretation.

Since different types of LAP have different spectral signatures (Figure 2), it is theoretically possible to assess the dominant type of LAP using our SIP measurements. With this aim in mind, we compute the relative optical impacts of dust and BC within this eqBC concentration. The fraction of total LAP absorption caused by dust ($\eta$) is computed as follows:

$$\eta = \frac{\psi(c_{\mathrm{dust}})}{c_{\mathrm{eqBC}}} \tag{2}$$

### 3.2 Selection of homogeneous layers in SIP measurement

Following the radiative transfer theory in a homogeneous layer far from any interface, the intensity $I(z, \lambda)$ decreases exponentially with depth $z$. This writes:

$$I(z, \lambda) = I(z_0, \lambda)\mathrm{e}^{-k_e(\lambda)(z - z0)}, \tag{3}$$

where $k_e(\lambda)$ is the Asymptotic Flux Extinction Coefficient (AFEC) expressed in m$^{-1}$. Simpson et al. (2002) explain that this equation is only applicable in the asymptotic region, the region where light is only diffuse and where the ground absorption has no influence. For this reason the 7 uppermost centimeters of the snowpack and the profiles acquired in shallow snowpacks (less than 50 cm) are discarded from our analysis.



For a homogeneous layer in the asymptotic region, the AFEC can hence be computed as the gradient of the log-radiance (logarithm of the irradiance) in the layer. However, Picard et al. (2016) have shown that the rod of SOLEXS can disturb the gradient of the log-radiance in the first centimeters around a transition between two layers of different scattering properties. For this reason, only homogeneous layers of the snowpack thicker than 3 cm can be exploited. Following the approach of Warren et al. (2006) and Picard et al. (2016), we visually determine zones having homogeneous properties based on the linearity of the log-radiance in the asymptotic region. We refer to those vertical layers with homogeneous properties as Zones Of Interest (ZOI). In total, we identified 100 ZOI over the 26 SIP measured over both seasons. Figure 3a shows an example of selected ZOI.

### 3.3 Asymptotic flux exctinction coefficient estimation

For every ZOI, we estimate the AFEC with a least-square linear regression of the log-radiance versus depth, based on Equation 3. To deal with the spectrometer noise for wavelengths where the signal is the weakest, the procedure to compute the AFEC for a specific ZOI is as follows:

1. For a given wavelength $\lambda$, if any $I(z,\lambda) \leqslant 0$, the AFEC is not computed.

2. The AFEC is computed for all remaining wavelengths as a linear regression of the log-radiance. Nevertheless the computed AFEC is often affected by SIP measurement noise for the largest wavelengths. To address this issue, the AFEC is decomposed into signal and noise. The signal is calculated by applying a convolution filter with a period of 11 nm on the raw estimate, and the noise is calculated as the difference between the raw AFEC and the filtered one.

3. The signal-to-noise ratio of the AFEC is estimated as the ratio between the average signal and the average noise over a window of 30 nm at the higher range of the spectrum. If this ratio is lower than 15, the AFEC in this range is discarded. If the signal-to-noise ratio is still lower than 15 in the next 30 nm, the last step is repeated. It should be noted that the signal-to-noise ratio is constantly higher than 15 at the lower range of the spectrum, i.e. around 350 nm. Figure 4 shows the selected maximum wavelength as a function of the bottom depth of the ZOI. Overall, the computation window varies between [350-680] nm and [350-944] nm with in general wider ranges at shallower depths. The maximum wavelength decreases with depth since the absorption of ice increases with wavelength. The relation is not deterministic because the available energy at a given depth also depends on the illumination conditions and on snow properties at the time of the measurement.

Figure 3b shows an instance of the spectral AFEC computation obtained for a ZOI before and after applying the convolution filter. For more clarity, the AFEC estimated from SIP measurements will be referred as "measured AFEC" in the following.

### 3.4 LAP retrieval algorithm

#### 3.4.1 Theory

The spectral AFEC ($k_e(\lambda)$) is related to snow single scattering properties (Wiscombe and Warren, 1980). Following Libois et al. (2013), under the delta-Eddington approximation (Joseph et al., 1976), for medias and wavelengths where scattering is much



stronger than absorption, the AFEC can be expressed as

$$k_e(\lambda) \approx \sigma_e \sqrt{3(1 - g\omega(\lambda))(1 - \omega(\lambda))}, \tag{4}$$

where $\sigma_e$ (m$^{-1}$), $g$ and $\omega$ are the extinction coefficient, the asymmetry factor and the single scattering albedo respectively. This equation applies, among others, to snow in the wavelength range targeted by this study (350-950 nm) where snow is strongly

scattering. Kokhanovsky and Zege (2004)'s theory for pure snow shows that for convex crystals :

$$\sigma_e = \frac{\rho SSA}{2}, \tag{5}$$

where SSA is the Snow Specific Surface Area (m$^2$ kg$^{-1}$; Legagneux et al., 2002). It can be expressed as SSA=$\frac{S}{\rho_{\text{ice}}V}$, where S is the ice matrix surface (m$^2$) of a given volume of ice (V; m$^3$) and $\rho_{\text{ice}}$ is the density of ice equal to 917 kg m$^{-3}$. This theory also shows that:

$$\sigma_a = \frac{\rho B \gamma_{\text{ice}}(\lambda)}{\rho_{\text{ice}}} \tag{6}$$

with $\sigma_a$ (m$^{-1}$) the absorption coefficient due to ice. The term $\gamma(\lambda)$ (m$^{-1}$) is related to the imaginary part of ice refractive index n$_i(\lambda)$ as follows:

$$\gamma_{\text{ice}}(\lambda) = \frac{4\pi n_i(\lambda)}{\lambda}. \tag{7}$$

It follows:

$$(1 - \omega)(\lambda) = \frac{\sigma_a}{\sigma_e} = \frac{2B\gamma_{\text{ice}}(\lambda)}{\rho_{\text{ice}}SSA}, \tag{8}$$

where $\rho$ is the snow density (kg m$^{-3}$). In the case of snow containing LAPs, assuming that scattering is only due to the ice air interfaces (Libois et al., 2013), Equation 8 can thus be written as:

$$(1 - \omega)(\lambda) = \frac{\sigma_a + \sigma_{a,LAP}}{\sigma_e} \tag{9}$$

with $\sigma_{a,LAP}$ the absorption coefficients due to LAPs. Assuming external mixing, $\sigma_{a,LAP}$ is expressed as:

$$\sigma_{a,LAP} = \sum_i MAE_i(\lambda)\rho_i = \rho \sum_i MAE_i(\lambda)c_i, \tag{10}$$

where $i$ runs over the different types of LAPs present in snow. For each LAP type $i$, $MAE_i$ is the Mass Absorption Efficiency (m$^2$ kg$^{-1}$; e.g., Caponi et al., 2017), $\rho_i$ is the mass concentration (kg m$^{-3}$) and $c_i$ the mass fraction in kg kg$^{-1}$. Equation 9 yields:

$$(1 - \omega)(\lambda) = \frac{2}{SSA} \left( \frac{B\gamma_{\text{ice}}(\lambda)}{\rho_{\text{ice}}} + \sum_i MAE_i(\lambda)c_i \right) \tag{11}$$





and finally:

$$k_e(\lambda) \approx \sqrt{\frac{3(1-g)}{2} \rho^2 SSA \left( \frac{B\gamma_{\text{ice}}(\lambda)}{\rho_{\text{ice}}} + \sum_i MAE_i(\lambda)c_i \right)}. \tag{12}$$

The interesting feature of this equation is that the spectral dependence of the AFEC comes only from two terms, $\gamma_{ice}(\lambda)$ and $MAE_i(\lambda)$ of the different types of LAPs. Figure 2 represents the spectral dependence of $\sigma_{a,snow}$, $\sigma_{a,dust}$ and $\sigma_{a,BC}$. As their three spectral signatures are remarkably different in the visible range, it is theoretically possible to separate the absorption due to ice and that due to each type of LAP.

### 3.4.2 Algorithm

In order to exploit Equation 12 to retrieve LAP concentrations from measured AFEC, several assumptions have to be made.

- The refractive index of ice is known and is taken from the most recent estimate (Picard et al., 2016).

- The types of LAP present in the snowpack are known. Here we assume two types: BC and dust without distinction within these categories.

- The Mass Absorption Efficiency (MAE) of these LAPs is known.

    - For BC it is derived from the constant BC refractive index advised by Bond and Bergstrom (2006) i.e $m = 1.91 - 0.79i$. As in the study of Hadley and Kirchstetter (2012), BC density is scaled to obtain a MAE of 11.25 m² g$^{-1}$ at 550 nm (11 m² g$^{-1}$ in their study), which is an intermediate value between fresh BC (around 7.5 m² g$^{-1}$ at 550 nm) and internally mixed aged BC (up to 15 m² g$^{-1}$ at 550 nm).

    - One of the prevailing dust source region for the Alps is the Saharan desert (Di Mauro et al., 2018). Consequently, the MAE of dust was set according to the values found in Caponi et al. (2017) for Libyan dust. The value advised for particles with a diameter inferior to 2.5 $\mu$m (PM 2.5) was chosen consistently with our chemical size distribution measurements.

- The snow shape parameters B and g are constant over time and for all types of snow. The enhancement parameter B is set to 1.6 and the asymmetry factor g is set to 0.85, considered to be good approximations to describe all type of snow (Libois et al., 2014).

Under these assumptions, the unknowns of the retrieval problem are BC concentration ($c_{BC}$), dust concentration ($c_{dust}$) and $\rho^2 SSA$. As $\rho^2 SSA$ is not an intuitive measure, we inject the measured density in Equation 12 so that our third unknown becomes SSA. However, other choices are equally possible without any interference in the LAP retrievals. For instance, leaving both density and SSA as free parameters would not impact the LAP retrievals. For each ZOI, once the measured AFEC has been computed (Section 3.3), a non-linear optimisation on SSA, $c_{BC}$ and $c_{dust}$ in Equation 12 is then performed to minimise the mean square error over all valid wavelengths between the estimated and the measured AFEC. The optimal parameters of this

minimisation are our best estimates of $c_{BC}$, $c_{dust}$ and SSA. Figure 6 shows an example of comparisons between the estimated and measured AFEC for a specific ZOI.

In some rare cases, the estimated AFEC does not fit well the measured one resulting in a RMSE between the estimated and the filtered measured AFEC higher than 3 m$^{-1}$. In these cases, the ZOI is discarded (5 out of 100). Since the theory
described above does not account for the presence of liquid water, 16 ZOI containing liquid water are discarded, as we found this has a great influence on SIP measurements that is not yet understood. For the 79 remaining ZOI, 55 have concomitant chemical measurements. Figure 7 shows the comparison between the retrieval algorithm on a specific SIP measurement and the corresponding snowpit measurements for a given field day.

In order to test the sensitivity of the method to the different modelling assumptions, numerical sensitivity analyses were
performed. The impact on LAP estimation is calculated by varying each parameter within its range of uncertainty, keeping the other parameters unchanged. The impact of the different modelling assumptions is discussed in Sections 4.2 and 4.3.

A scheme synthesizing the whole methodology found in this Section is presented in Figure 5.

## 4   Results

### 4.1   LAP estimation with optimal parameters

Figure 8 compares the LAP concentrations estimated from the SIP measurements to the snowpit chemical measurements under the assumptions detailed in Section 3.4.2. The symbols correspond to the 55 ZOI for which corresponding chemical measurements are available. The horizontal error bars correspond to the measurement uncertainties described in Section 2. The color of the symbols indicates the contribution of dust to the total LAP impact according to chemical measurements ($\eta_{\mathrm{mes}}$ from Equation 2). The size of the symbols corresponds to the span of wavelengths used for the estimation, in other words the size of
the symbols increases with the maximum wavelength on which the retrieval algorithm is applied. The wavelength range of the estimation and the $\eta_{\mathrm{mes}}$ do not impact the eqBC retrieval.

This figure has two important implications; first, the retrieval method is not sensitive to LAP amounts lower than 5 ng g$^{-1}$ eqBC, which may seem disappointing because it greatly reduces the number of validation points, nevertheless it was expected that the algorithm has a limit of sensitivity. 5 ng g$^{-1}$ is in line with the observations of Picard et al. (2016) in Antarctica. For this
reason all the points with a measured eqrBC concentration lower than 5 ng g$^{-1}$ are discarded from the statistics presented in the following. Second, the algorithm shows a sensitivity in the range 5–60 ng g$^{-1}$. Indeed, the correlation in this range has a r$^2$ of 0.74 in spite of a significant bias of 15.7 ng g$^{-1}$ eqBC. The main purpose of the following is thus to investigate the cause of this bias by focusing on the snow layers with sufficient LAPs to be detected.

### 4.2   Impact of LAP properties

Figure 9 shows how the algorithm is impacted by uncertainties on LAP optical properties. The symbols are the same as in Figure 8 with additional vertical error bars corresponding to the retrieval uncertainties caused by uncertainties on LAP MAE.





The uncertainty on BC MAE is considered to be bounded by the two extreme values found in Hadley and Kirchstetter (2012) (7.5 and 15 $m^2$ $g^{-1}$ at 550 nm). This uncertainty induces a -26.6%, +46.6% uncertainty on our BC estimation, shown by the vertical bars in Figure 9a. Figure 9b shows the impact of dust MAE, considered as follows. Caponi et al. (2017) suggest that for dust particles smaller than 2.5 $\mu$m (PM2.5), which is the major dust type in regard of measured size distribution, dust MAE at

407 nm is between 0.071 and 0.127 $m^2$ $g^{-1}$ (0.103 for Figure 8). The variations of dust MAE are assumed to span this range, inducing an asymmetric uncertainty of -19%, +45.1 % on dust estimation. The impact of changes in the spectral signature of dust absorption is not included here but is discussed in Section 5.3.

## 4.3   Impact of snow physical parameters

Both density and SSA were measured in the field. These measurements are not necessary to apply our LAP retrieval algorithm

but it is interesting to check if the SSA leading to the correct absorption is consistent with the measured SSA. Figure 10 shows the estimated SSA compared to the measured SSA for the 68 ZOI previously selected for which SSA measurements are available. The horizontal error bars correspond to uncertainties on SSA measurements described in Section 2. Following Equation 12 the AFEC is proportional to $\sqrt{\rho^2 SSA}$. For a given AFEC, the 5% uncertainty on density measurements thus introduces an asymmetric uncertainty of -9.1%, +11.1 % on SSA estimation (vertical error bars). There is a correlation between estimated

and measured SSA with a $r^2$ of 0.73 and no significant bias indicating that SSA variations are well captured by our retrieval algorithm. This result indicates that our LAP retrieval algorithm coupled with density profile measurements can also bring a relatively accurate estimation of SSA.

The SSA measurements are obtained from NIR reflectance based on the hypothesis that the shape parameters B and g, from Equation 12, are related by $\frac{B}{1-g}$=10.7. This value is considered to be good approximation to describe all types of snow (Gallet

et al., 2009; Arnaud et al., 2011; Libois et al., 2014). However the enhancement parameter B and the asymmetry factor g are expected to vary during snow metamorphism (Libois et al., 2013; Kokhanovsky and Zege, 2004) but their evolution is poorly documented. Libois et al. (2013) quantified the theoretical variations of B, g for different geometric shapes highlighting a high variability of these parameters. Under the constraint $\frac{B}{1-g}$=10.7, B and g can still vary according to grain shape leading to potential variations of B(1-g) affecting our retrieval method. To account for these variations we selected extreme B and g values

respecting this constraint based on Figure 1 in Libois et al. (2013). Figure 11 illustrates the impact of B and g variations on the retrieval of LAP concentrations. The numerical analysis shows that the relative impact of shape parameter variations on the estimation is independent of the SSA and LAP concentration values. Overall, B variations lead to -10%, +25% uncertainty on impurity estimation. The variations of g do not impact LAP retrievals.

Uncertainties on the refractive index of ice may also slightly impact our results. The refractive index proposed by Warren et al.

(2006), being lower than the one of Picard et al. (2016) used in this study, would lead to less absorbing ice in the spectral range 400-600 nm, implying higher estimates of LAP concentrations. This would increase the bias observed in Figure 8 of around 1 ng $g^{-1}$ eqBC (estimate not shown). The impact is low regarding other sources of uncertainties and is not further explored.



### 4.4 SIP spectral information

Figure 14 illustrates the impact of considering only one type of LAP (BC here) instead of two in our retrieval algorithm. In a first example of ZOI (Figure 14a), the absorption is dominated by BC and both retrievals have similar performances considering dust or not. In a second ZOI (Figure 14b), dust is clearly the dominant absorber and has been measured with a concentration of about

13 $\mu$g g$^{-1}$. In this case the estimated AFEC from the retrieval algorithm does not reproduce the measured one by accounting only for BC. The presence of a different LAP type with an higher Angstorm exponent, dust here, is necessary to explain the spectral signature of the AFEC in the visible.

In order to investigate if finer information on the LAP prevailing type can be retrieved, the estimated contribution of dust to the total LAP impact ($\eta_{est}$ from Equation 2) is shown in Figure 12 and compared to the measured dust proportion over the 14

ZOI with a measured eqrBC concentration higher than 5 ng g$^{-1}$. The retrieval method is sensitive to the type of LAP present in the snowpack with a low $\eta_{est}$ when BC dominates (median value of 0.1) and higher values of $\eta_{est}$ when dust dominates (median value of 0.6). At this stage of development, only these cases can be distinguished but not quantitative measure of the relative contribution. Moreover, the few number of validation points and the presence of dust in most of the ZOI where measured eqrBC concentration are higher than 5 ng g$^{-1}$ make it difficult to draw a reliable conclusion and this result has to be taken with care.

### 4.5 Impact of liquid water

Figure 13 shows an example of application of our method to a ZOI containing liquid water. The estimated LAP concentration is one order of magnitude higher than the measured one. A similar phenomenon has been systematically observed in the 16 ZOI in which liquid water is present, which is why they were discarded from Figure 8. The measured AFEC is abnormally high between 350 and 700 nm in regard to the measured LAP concentration, causing a strong overestimation of LAP concentrations.

Further investigation is needed to understand the cause, but the consequence is that information about LAP cannot be retrieved in the presence of liquid water with our methodology.

## 5   Discussion

### 5.1   Discrepancy between measured LAP concentrations and induced absorption

Figure 8 shows a correlation between LAP concentrations estimated from SIP and chemically measured ones, which suggests

that easy measurements of the optical impact of LAP may be possible in the future. However, there is still a strong uncertainty and clear positive bias between impurity contents estimated from the measured AFEC and the measured ones. Most of the uncertainties may be due to uncaught variations of LAP optical properties (Figure 9) and snow physical parameters (Figure 11), which is illustrated by the fact that, when subtracting the 15.7 ng g$^{-1}$ eqBC positive bias, all measured LAP concentrations higher than 5 ng g$^{-1}$ are within the range of uncertainty of the retrieval.





Even with the aforementioned uncertainties, the eqBC concentration retrieved in some ZOI does not match the measurements. This suggests that for a given measured concentration of LAP, the radiative impact induced on snow absorption is too low. We see three potential explanations for this:

- A problem in our SIP measurements cannot be excluded: the disturbance caused by the fiber rod is discussed in Section 5.3. However, France et al. (2012) also noticed that BC and humic-like substances estimated from SIP measurements was abnormally higher than the one measured at the same location during the OASIS campaign (Voisin et al., 2012). As they used a different measurement technique, the bias is probably not due to the measurement technique.

- The problem may come from chemical measurements of LAP in snow. The bias observed here could be explained by a systematic underestimation of chemically measured LAP concentrations in snow as suggested in Schwarz et al. (2012) for BC. The particle size of BC was found to be larger in snow than in the atmosphere (Schwarz et al., 2013), which may lead to the underestimation of measured rBC concentrations because the larger sizes are not detected by the SP$^2$. This is partly accounted for in the chemical data processing, but implicitly depends on having an external calibrant with a size distribution close enough to that of the actual BC in snow. The calibrant chosen here (Wendl et al., 2014) reduces the underestimation to a minimum, without excluding it totally. As for dust, our measurements present potential biases in both directions: some sedimentation during the handling of the sample is always possible, although our protocol was designed to minimize the risk (the samples are gently shaken while waiting for analysis). On the opposite, we assume that all the measured insoluble particles above 0.6 μm are light absorbing dust, which may lead to overestimating the dust concentrations: some of the higher size particles might be non light absorbing dust (such as quartz, or calcite). It is thus unlikely that the whole bias can be explained with this sole hypothesis.

- This suggests a third hypothesis: our LAP MAE uncertainty estimation does not span across a wide enough range. LAP enhancement absorption when deposited in snow may be underestimated due to LAP-snow physico-chemical interactions. This absorption enhancement is often attributed to internal mixing of LAP in snow (e.g., Flanner et al., 2012) but might be partly due to other physical processes such as coating of LAP particles, which remains poorly investigated in snow despite its strong impact on LAP absorption in the atmosphere (Moffet and Prather, 2009).

## 5.2 Impact of water

Figure 13 reveals a strong unexplained extinction enhancement in the visible if liquid water is present. This phenomenon has been observed in all ZOI containing liquid water. In some case, an anormally high extinction is also observed in the NIR part of the spectra. We propose two possible explanations. First, liquid water may enhance LAP absorption due to chemical or optical interactions, having a consequent impact on light extinction. For instance, the inclusion of externally mixed LAP in liquid water might cause a lensing effect increasing consequently the MAE of the present LAP. Mikhailov et al. (2006) suggests that soot-water drop aggregates can enhance absorption of the soot particle by a factor up to 3 compared to the sole particle. This could explain the extinction enhancement observed on the AFEC of layers containing liquid water observed in Section 4.5. Second, it might be due to experimental problem as the hole in which the fiber is inserted is made of air. In case of a very wet





snowpack, inserting the metal rod into the snowpack may create a water lens around the rod, creating an additional air/water interface around the optic fiber. This might perturb the SIP measurement and in turn the AFEC.

Libois et al. (2013) tried to determine the value of the shape parameter B from data in the literature based on AFEC estimations with concomitant reflectance measurements. The two values of B retrieved for snowpacks containing liquid water
are questionable, which may originate from the same issue observed in our study.

### 5.3   Additional sources of error in the measurements

**Impact of the rod**

The uncertainties affecting SIP measurements with the technique used in this study have been assessed in Picard et al. (2016) for pristine snow. As the measurement protocol advised by their study has been strictly followed in our SIP measurements, they
suggest that uncertainties are expected to be less than 1 ng g$^{-1}$ eqBC. In the study of Picard et al. (2016) the impact of the rod is significant below 500 nanometers for extremely small amounts of LAP (about 1 ng g$^{-1}$), i.e AFEC around 5 m$^{-1}$. Despite the higher level of LAP in Alpine than Antarctic snow, we also observe on our SIP measurements some non-physical behavior at the transitions between the homogeneous layers that Picard et al. (2016) estimated to be only possible for pristine snow. This includes for instance short zones in the profiles with increasing radiance with depth (e.g. in Figure 3a around 32 cm depth). This
clearly break the radiative transfer theory for 1D plane parallel media, and might explain a part of the uncertainties of the present method, especially for low impurity contents. Figure 8 clearly shows that retrieval under 5 ng g$^{-1}$ is not possible, spotlighting a strong dispersion of the LAP estimation which might be partly explained by the impact of the rod.

**Presence of other LAPs**

Here, we consider BC and dust to be the only absorbers present in the snowpack. The presence of other LAP types in the
snowpack uncaptured by our chemical measurements might explain a part of the bias between optically retrieved and chemically measured LAP concentrations observed in this study. For instance, Organic Carbon (OC) may play an important role in snowpack absorption (Lin et al., 2014; Wang et al., 2019). However, the peak absorption of OC is located between 350 and 400 nm and the impact at wavelengths higher than 400 nm is expected to be limited (Chen and Baker, 2010). It is hence expected to have a low impact on this work though more impacting for photochemistry in the UV.

**Spectral signature of LAPs**

In addition to the uncertainty on absorption efficiency of LAPs discussed in Section 4.2, the spectral signature of LAP absorption can also vary. For BC, the absorption Angstorm exponent is around 1 and is not expected to vary significantly (Bond and Bergstrom, 2006). On the contrary dust Angstorm exponent can vary from 2 to 5 depending on the source and size distribution of dust (e.g., Caponi et al., 2017 ) and is assumed to be to 4.1 in the present study (Libyan dust). Considering a different Angstorm
exponent for dust would not impact significantly the eqBC retrieval but would modify the partition of LAP impact between dust and BC.



## 6  Conclusion

This paper presents a unique dataset including two seasons of near-weekly surveys of snow physical properties (SSA, density) associated with measurements of spectral irradiance profiles (SIP). The asymptotic flux extinction coefficient (AFEC) is estimated from SIP measurement in homogeneous layers of the snowpack in the visible and NIR. In each layer, we determine

the optimal LAP concentration explaining the measured spectral AFEC using the asymptotic approximation of the radiative transfer theory (AART; Kokhanovsky and Zege, 2004). Through a comparison of these optimal LAP concentrations with chemical LAP measurements, we demonstrate that valuable information on properties of LAP in snow can be estimated from SIP measurements.

For the first time, we compare the spectral signal of LAP with snow extinction and chemical analyses of LAP concentrations.

For now, the limit of sensitivity of our method is around 5 ng g$^{-1}$ and smaller concentrations can not be detected. For higher concentrations, we highlight a correlation between estimated eqBC concentrations and measured ones (r$^2$=0.74 ). We also demonstrate that the spectral information of LAP can be retrieved from SIP measurements. It is possible to determine the prevailing type of LAP present in a layer based on its spectral signature. However the reliability of this method is relatively poor for now. Our results suggest that LAP absorption is enhanced in layers containing liquid water, where our method does

not apply. This might come from the formation of LAP-water aggregates as described in Mikhailov et al. (2006) or from a measurement artefact. The method proposed also gives valuable information on snow physical properties which are left as a free parameter. We verify that the estimated snow properties are consistent with measurements, demonstrating a good correlation between estimated SSA and in-situ SSA measured by NIR reflectometry (R$^2$=0.73 ).

This study is a promising first step to easily determine vertical profiles of LAP concentrations within the snowpack. However

the accuracy of our retrieval method is low and a marked positive bias of around 16 ng g$^{-1}$ eqBC is observed. The low accuracy is not surprising given the strong uncertainties of LAP absorption efficiency and of snow physical parameters in the modelling. Nevertheless, the cause of the bias can not be explained assuming reasonable uncertainties in the modelling parameters. The potential causes of the bias discussed, raise different issues: SIP measurement uncertainties, chemical measurement uncertainties or underestimation of LAP in snow absorption enhancement due to interactions between LAP and snow. The bias between LAP

radiative impact and chemical measurements is challenging to address owing to several reasons. Firstly, chemical measurements in snow are time consuming and are affected by many uncertainties such as the dependence on the size distribution of the particles or the nebulisation biases. Secondly, LAP optical properties in snow are highly variable and their evolution is poorly understood. The mixing state, the coating, or the presence of liquid water affect the absorption efficiency of LAP and need to be further investigated. Thirdly, uncertainties on snow microstructure introduce high uncertainties in the retrieval method. Using

Monte Carlo ray tracing on real micro-tomography snow samples might be a way to better understand these parameters in link with snow metamorphism (Kaempfer et al., 2007). Any advances on one of these points are expected to lower the uncertainties affecting LAP absorption efficiency in snow and in turn the method presented here. As SIP measurements are much faster than manually collecting profiles of chemical measurements, our method could be attractive as an alternative to extract vertically resolved information on LAP concentrations in snow.



**Author contributions**

F. Tuzet led the study and was in charge of the field measurement campaign over the two seasons. F. Tuzet along with M. Dumont and G. Picard performed the major part of the data anlysis. L. Arnaud and G. Picard developed the SOLEXS instrument and the code library to process the measured spectral irradiance profile data. D. Voisin supervised the chemical measurement
analysis. M. Lamare, F. Larue and J. Revuelto had a major contribution in the field data acquisition. All authors contributed to the manuscript.

**Acknowledgments**

CNRM/CEN and IGE are part of Labex OSUG@2020 (investissement d'avenir - ANR10 LABX56). This study was supported by the ANR programs 1-JS56-005-01 MONISNOW and ANR-16-CE01-0006 EBONI; the INSR/LEFE projects BON and
ASSURANCE; the Ecole Doctorale SDU2E of Toulouse. This research was at least partially supported by Lautaret Garden-UMS 3370(Univ.Grenoble Alpes, CNRS, SAJF, 38000 Grenoble, France), member of AnaEE-France (ANR-11-INBS-0001AnaEE-Services, Investissements d'Avenir frame) and of the eLTER-Europe network (Univ. Grenoble Alpes, CNRS, LTSER Zone Atelier Alpes, 38000 Grenoble, France). The authors are grateful to LISA and PSI for chemical analysis of snow samples presented in this study. J. Revuelto is supported by a Post-doctoral Fellowship of the AXA research fund (Ref: CNRM 3.2.01/17).
The authors are also very grateful to all the people that helped perform the field measurements and chemical analyses: Bertrand Cluzet, Ines Ollivier, Clement Delcourt, Paul Billecocq, Natasha Bradford, Anthony Vella, Vincent Lucaire, François Besson, Matthieu Vernay, Frederic Flin, Céline Vargel and Mark Flanner. Finally, special thanks to Jacques Roulle for his commitment to fixing the field instruments, his help in the field, and his precious insight.

**Data and code availability**

The datasets analysed during this study and the code used to produce the figures are available from the corresponding author on request.




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





**Tables**

| Acronym | Full name |
|---------|-----------|
| SIP | Spectral Irradiance Profile |
| LAP | Light Absorbing Particle |
| AFEC | Asymptotic Flux Extinction Coefficient |
| SSA | Specific Surface Area |
| MAE | Mass Absorption Efficiency |
| ZOI | Zone Of Interest |
| BC | Black Carbon |
| rBC | refractory Black Carbon |
| eqBC | equivalent Black Carbon |
| eqrBC | equivalent refractory Black Carbon |
| NIR | Near InfraRed |

**Table 1.** Summary of the acronyms used in the present study.





## Figures

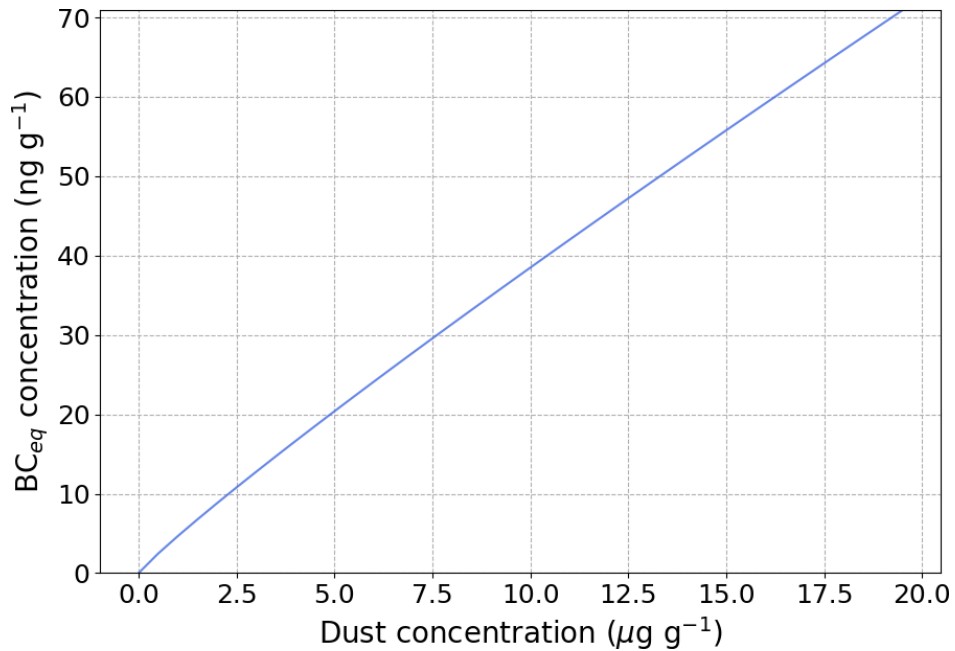

**Figure 1.** EqBC concentration corresponding to a given dust concentration with the model configuration (e.g.: B, g, LAP MAE) used in the present study.

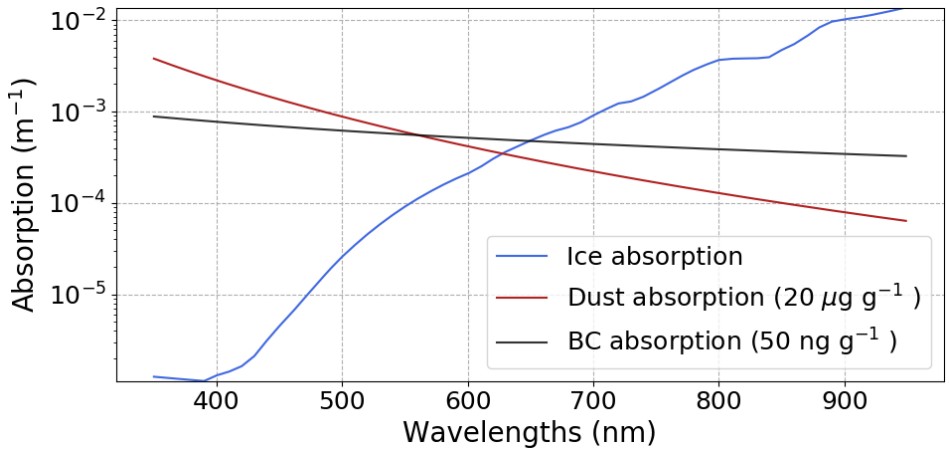

**Figure 2.** Spectral signature of the absorption coefficients $\sigma_a$ for ice and different types of LAPs.





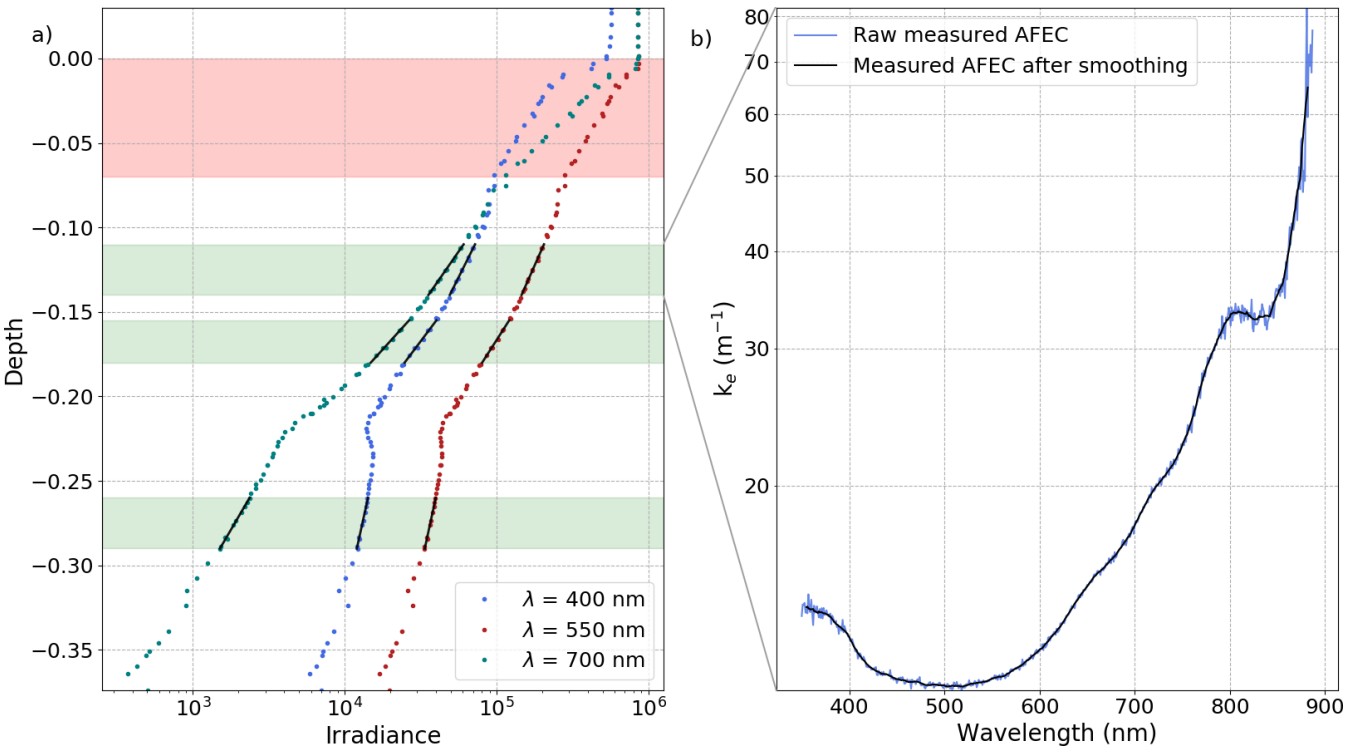

**Figure 3.** a) Irradiance as function of depth for selected wavelengths for SOLEXS profile 002 on 13/02/2018. Green shading shows the zones of interest (ZOI), which are homogeneous layers where the decrease in irradiance is visually linear on a logarithmic scale. The red shading corresponds to the part of the snowpack discarded due to the potential influence of direct light. b) Measured AFEC (blue curve) and filtered AFEC (black curve) as a function of wavelength. Note that the ordinate scale is logarithmic.

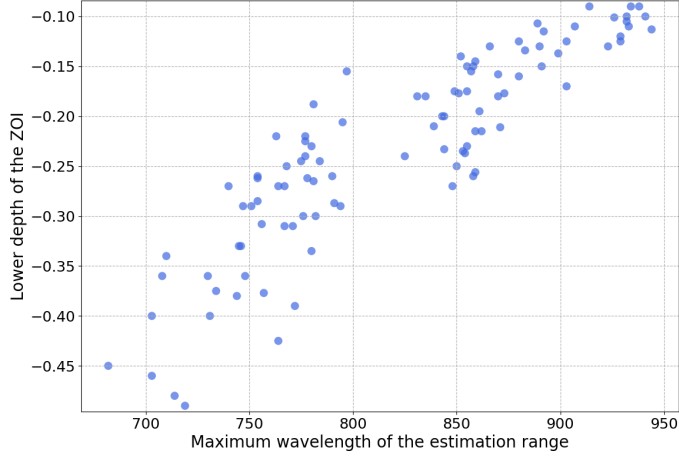

**Figure 4.** Upper limit of the spectral range where the AFEC estimation shows a signal to noise ratio over 15 for the whole dataset (100 ZOI).





**Figure 5.** Scheme synthesising the principle of the LAP retrieval method presented in Section 3.





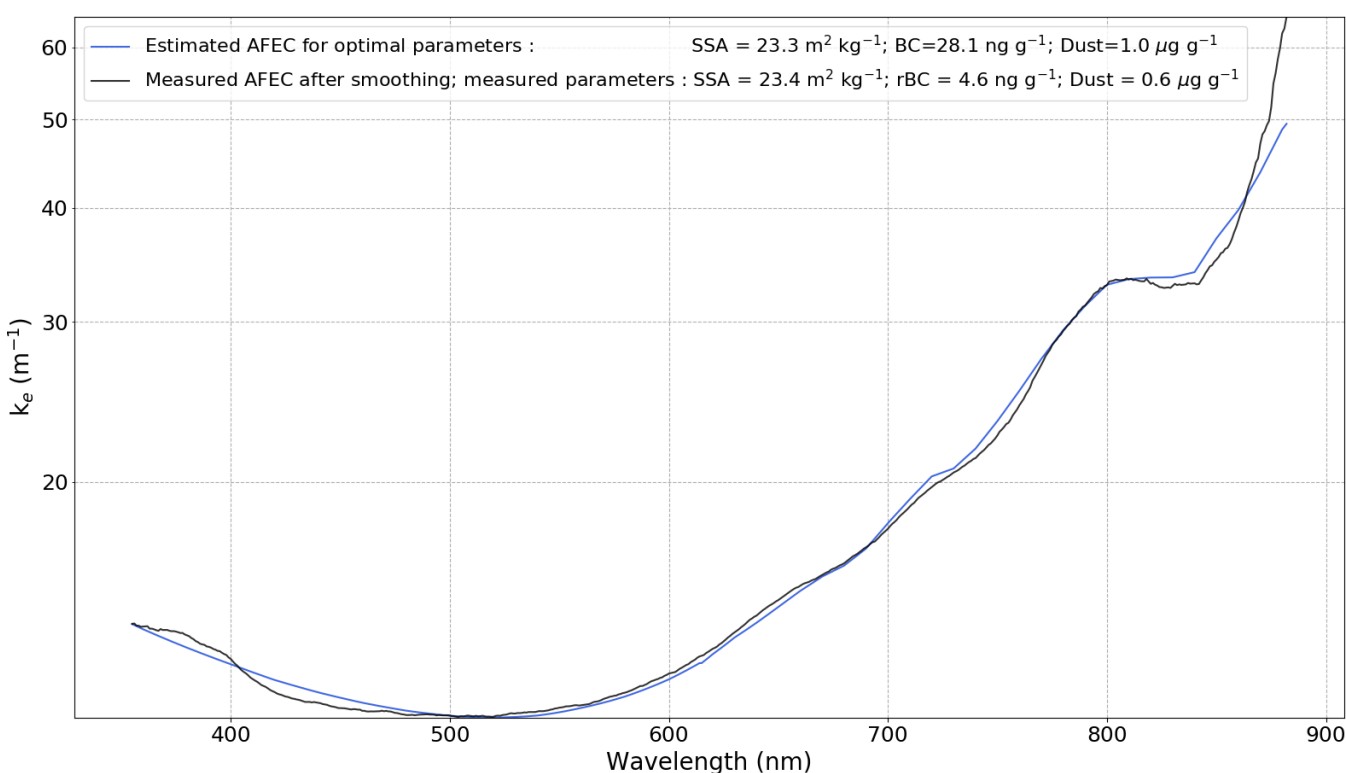

**Figure 6.** AFEC as a function of wavelength for the ZOI between 11 and 14 cm on the SOLEXS profile 002 on 13/02/2018. Measured AFEC after convolution filtering (black curve) is compared to estimated AFEC from Equation 12 with optimal parameters (blue curve).





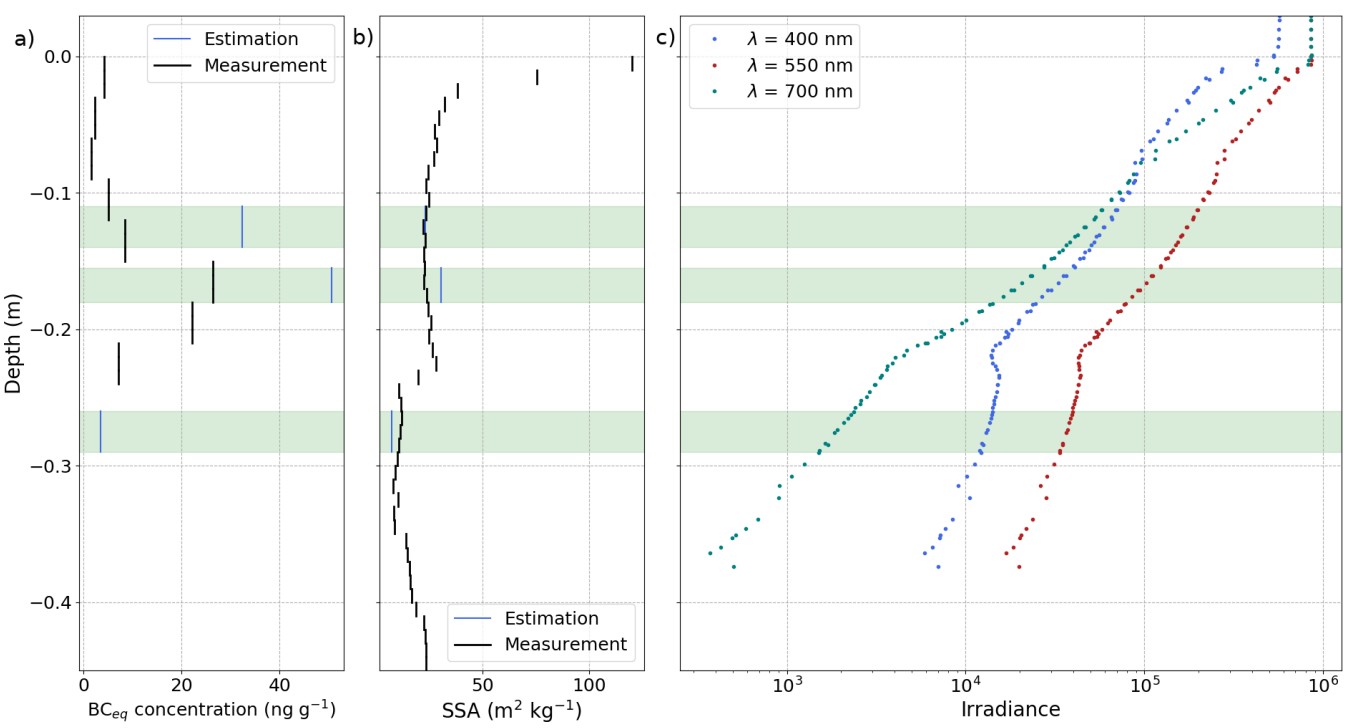

**Figure 7.** Comparison between snowpit measurements and estimated SSA and LAP concentrations for the SOLEXS profile 002 on 13/02/2018, green shading corresponds to the different ZOI of the profile. a) Vertical profile of eqBC concentration; measured (black) and estimated from AFEC optimisation on each ZOI (blue). b) Vertical profiles of SSA; measured (black) and estimated from AFEC optimisation on each ZOI (blue). c) SIP measurement from which AFEC has been derived.





**Figure 8.** Comparison between measured and estimated eqBC concentrations for all the ZOI with concomitant LAP measurements. The gray shading corresponds to the zone below the sensitivity limit of our method (i.e : 5 ng g$_{-1}$). The linear fit in dotted black line is computed for points where eqrBC measured concentration is higher than 5 ng g$^{-1}$. The color of the symbols corresponds to the proportion of LAP absorption coming from dust and their size is related to the maximum wavelength of the AFEC estimation.





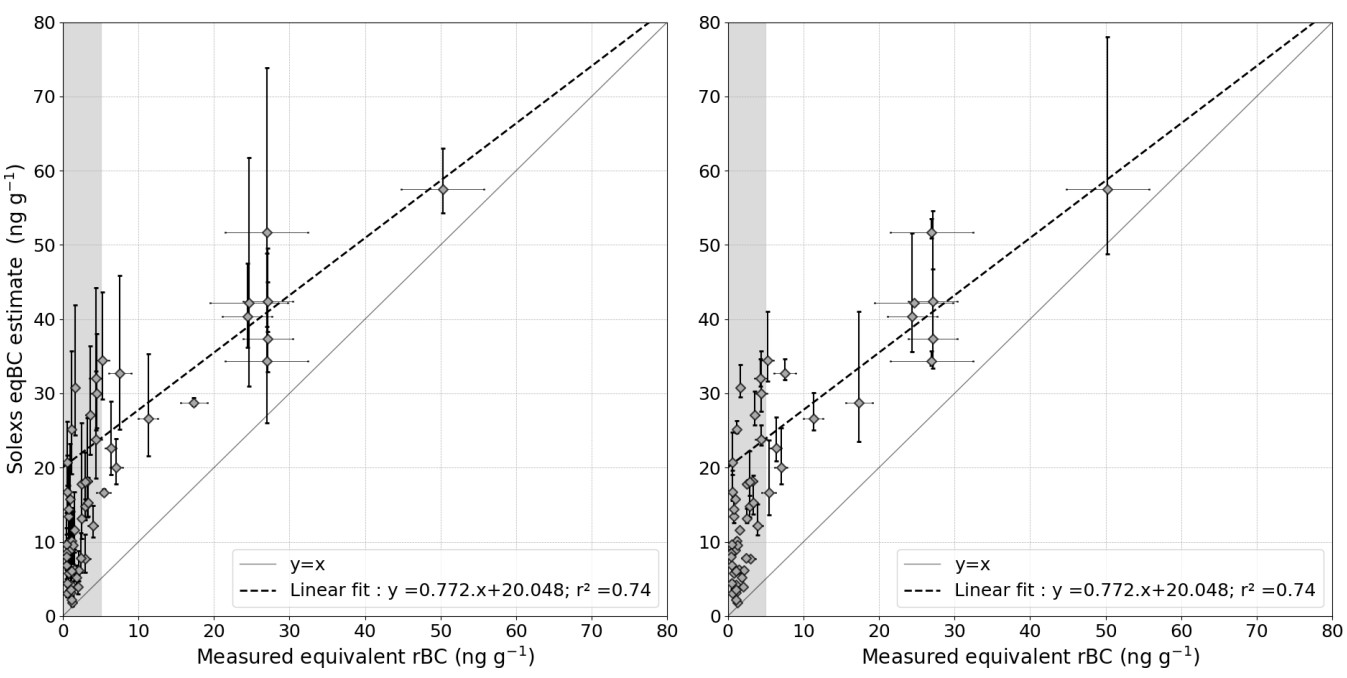

**Figure 9.** Comparison between measured and estimated eqBC concentrations for all the ZOI with concomitant LAP measurements. The gray shading corresponds to the zone below the sensitivity limit of our method (i.e : 5 ng g$^{-1}$). a) Error bars show how uncertainties on BC MAE affect LAP estimates. b) Error bars show how uncertainties on dust MAE affect LAP estimates.





**Figure 10.** Comparison between measured and estimated SSA for all the ZOI comporting SSA measurement. The color of the symbols corresponds to the proportion of LAP absorption coming from dust and sizes are related to the maximum wavelength at which the AFEC is estimated. Symbols are blue when chemical measurements are not available.



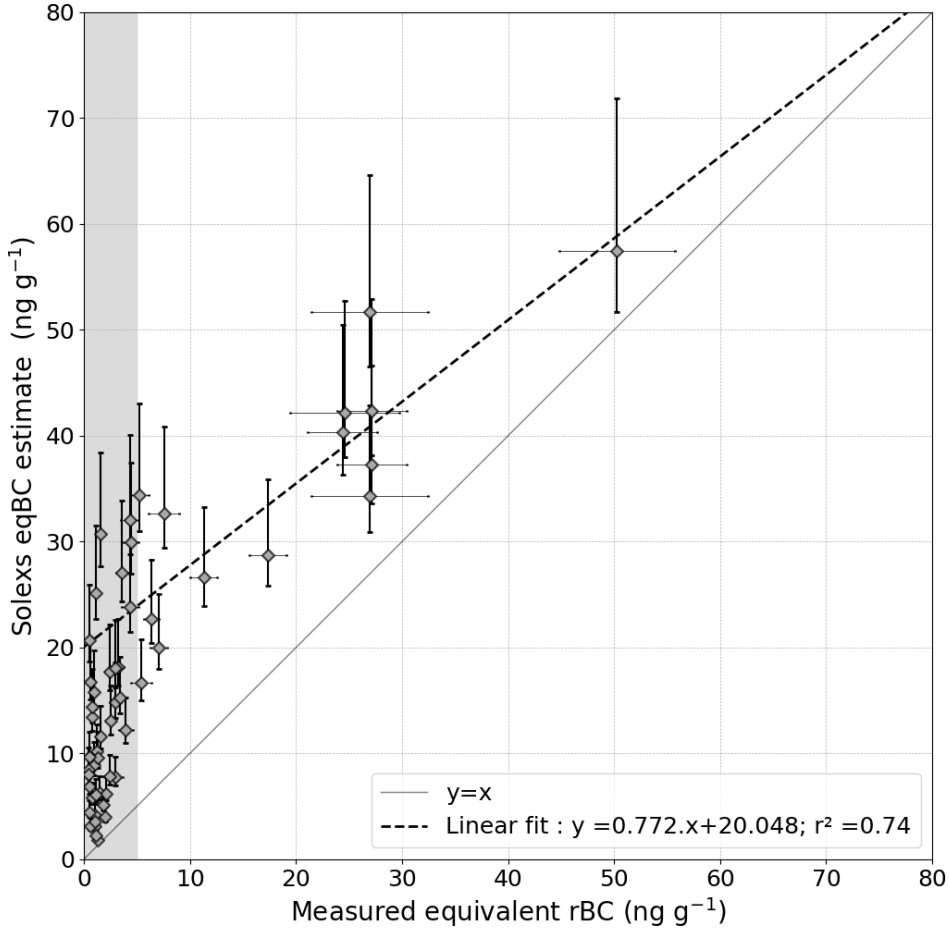

**Figure 11.** Comparison between measured and estimated eqBC concentrations for all the ZOI comporting chemical measurement. The gray shading corresponds to the zone below the sensitivity limit of our method (i.e : 5 ng $g_{-1}$). Error bars show how uncertainties on the enhancement parameter of ice B affect the LAP retrieval algorithm.







**Figure 12.** Comparison between measured and estimated proportion of LAP absorption coming from dust for all the ZOI with concomitant LAP measurements. The size of symbols corresponds to the measured eqBC concentration of the associated ZOI.



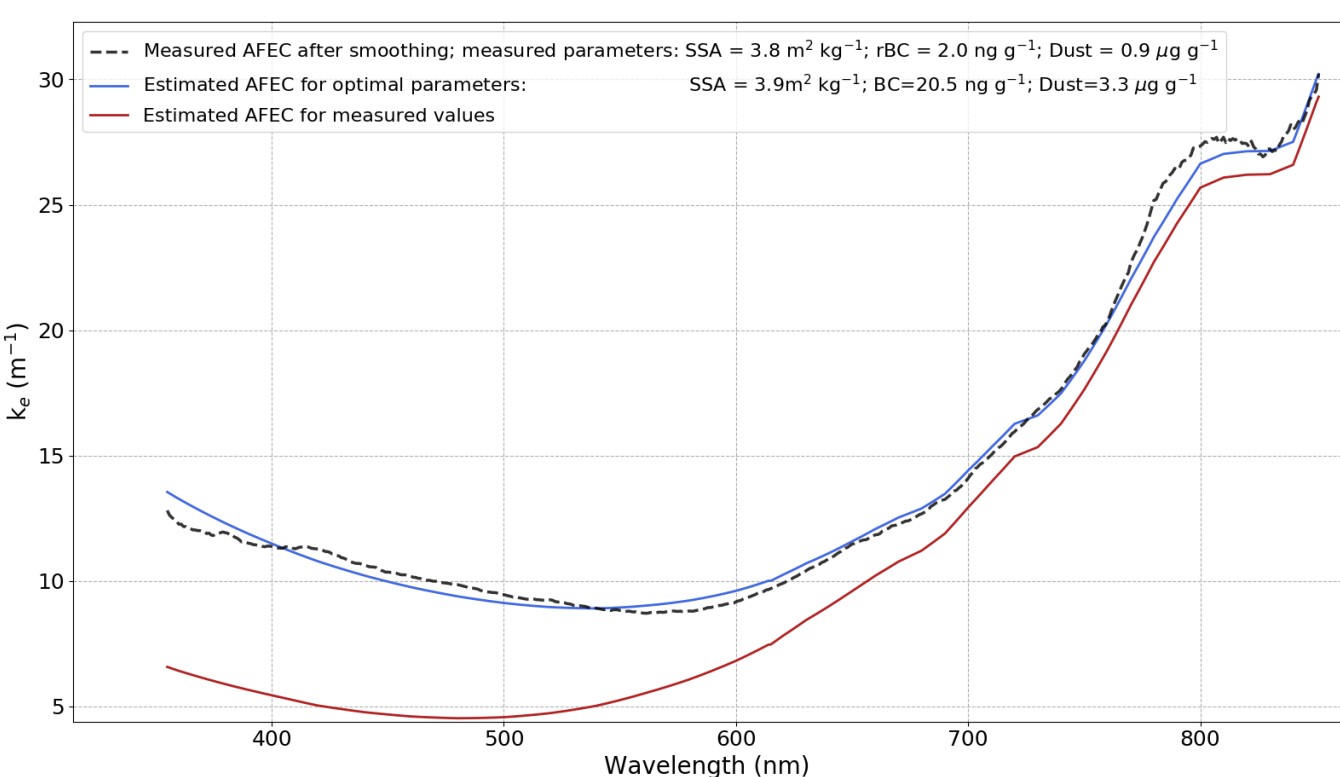

**Figure 13.** AFEC computation on a ZOI containing liquid water (between 15 and 18 cm on the SOLEXS profile 002 measured on 28/03/2017). The measured AFEC after filtering (black dotted line) is compared to the AFEC modeled using optimal parameters (blue curve) and to the AFEC modeled with measured values (red curves).





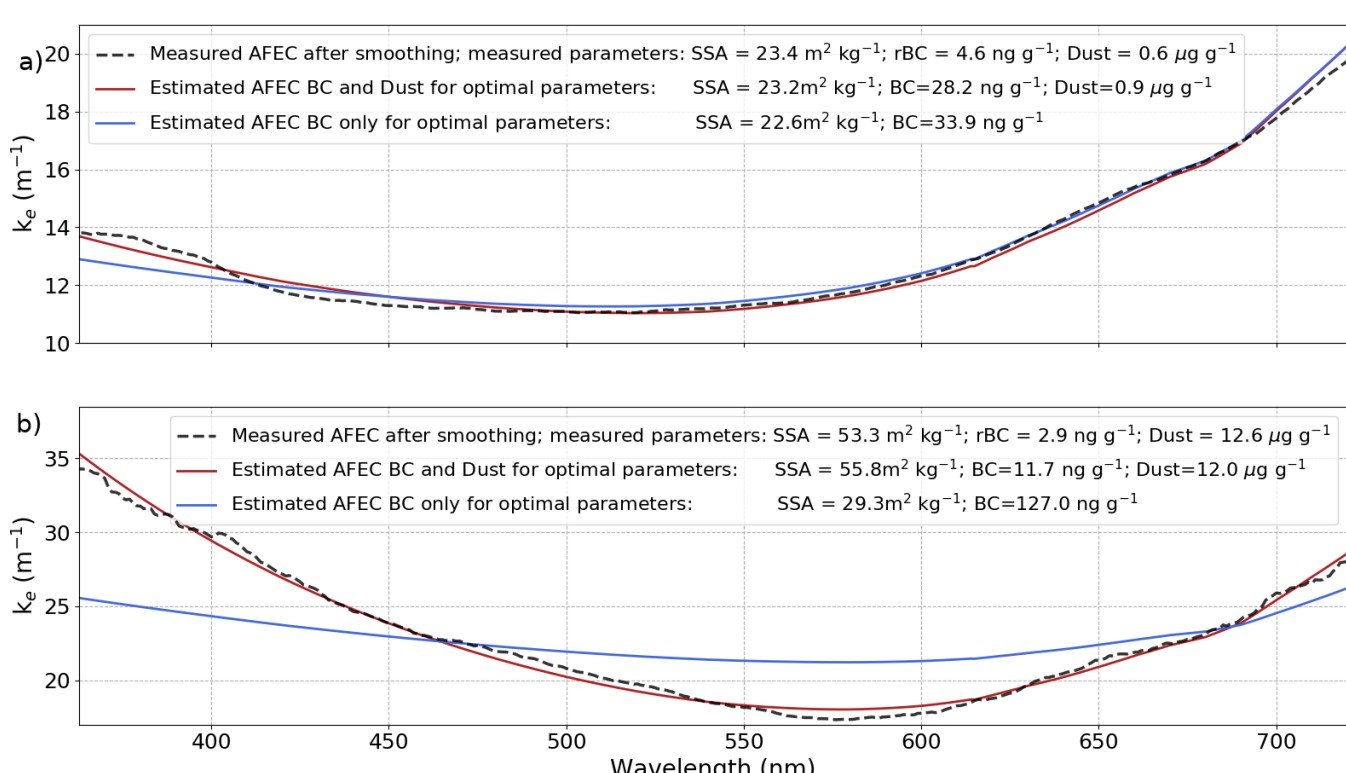

**Figure 14.** AFEC measured (black dotted line) and estimates considering both dust and BC (red curves) or considering only BC (blue curves) in the retrieval algorithm. a) On a ZOI located between 11 and 14 cm on the SOLEXS profile 002 measured on 23/02/2018, where LAP absorption is dominated by BC. b) On a ZOI located between 33 and 36 cm on the SOLEXS profile 004 measured on 01/10/2018, where LAP absorption is dominated by dust (around 10 $\mu g \ g^{-1}$ of dust measured).