# Peer review of "Influence of light absorbing particles on snow spectral irradiance profiles"

_The Cryosphere, 2019_

## Referee Comment (RC1) · Anonymous Referee #1 · 30 May 2019

**General comments**

This paper presents an original technique for estimating the concentration of light absorbing particles (LAP) in snow based on measurements of vertical profiles of spectral irradiance made at wavelengths from 350 to 950 nm.

The approach is based on values of the asymptotic flux extinction coefficient (AFEC) derived for homogeneous layers in snow. An optimization algorithm is developed that estimates the snow specific surface area (SSA) and concentratios of dust ($c_{dust}$) and black carbon ($c_{BC}$) based on the AFEC spectra. The inferred parameters are then compared with independent observations of SSA and equivalent-BC. It is shown that the estimated equivalent-BC concentration correlates quite well with that inferred from

chemical measurements, although only for concentrations larger than 5 ng g$^{-1}$. However, disturbingly, there is a substantial systematic bias of ≈16 ng g$^{-1}$ between the equivalent-BC concentrations inferred from the algorithm and the chemical measurements. Various sources of error are discussed, but they are able to explain the bias only partially.

The proposed method would allow for a relatively fast measurement of the LAP concentrations (compared to chemical analyses of snow samples), but at this stage, its attractiviness is reduced by the presence of a systematic bias that is not properly understood. Also, the applicability of the method is limited to homogeneous layers, and it can only be used at depths where the radiation field in snow is diffuse, and sufficiently far away from the underlying ground. Therefore, rather than providing a method of measuring LAPs that is ready to use at its present state, I view the research reported in this article as a "closure experiment" that probes our understanding of both the radiative transfer in snow with LAPs and of the measurements. Clearly, this understanding is still less than perfect.

Even though the proposed method does not yet work fully satisfactorily, I think this is an innovative and interesting study. I therefore recommend its publication in the Cryosphere, subject to the minor comments listed below.

**Specific comments**

1. p. 2, line 2: Can you specify what you mean with "near infrared"? Different definitions exist. Wikipedia mentions both 1.4 $\mu$m and 2.5 $\mu$m, while in atmospheric radiative transfer literature, NIR usually extends to 4 $\mu$m. It is well known that snow albedo is generally quite low at wavelengths larger than ≈1.4 $\mu$m.

2. p. 4, line 11: do you mean the onset of the snow season, or onset of the snow melt

season?

3. p. 4, lines 17–18: "taking precautions not to enlarge the hole". Even so, some light could leak deeper into snow through the small space of air between the rod and the snowpack. Did you eliminate this somehow?

4. p. 6, line 11: It would be helpful to show the spectral Mass Absorption Efficiencies (MAE) of BC and dust assumed in the computation of equivalent BC concentration in the conventional units ($m^2\,g^{-1}$). I can see that related information is given in Fig. 2, but it requires effort to do the conversion.

5. p. 6, Eq. (3): please define $z$ and $z_0$. Presumably $z$ is the depth, increasing downwards (even though in Figs. 3 and 7, the values are negative).

6. p. 6, line 27: Did you apply some quantitative criteria for the minimum distance between your zones-of-interest and ground?

7. p. 8, Eq. (6): please define $B$ (absorption enhancement parameter?).

8. p. 9, Eq. (12): Strictly speaking, $g$ and $B$ also depend on wavelength. An example of this can be seen in Figure 6a,c of Räisänen et al. (2015) for a few assumptions about snow grain shapes. (I think their $\xi$ parameter is equal to $B$ in this study). For the wavelengths of interest for the present study (350-950 nm), $B(1-g)$ might vary up to $\sim \pm 5\%$ as a function of wavelength (with largest values at the short wavelengths). I guess this would be insignificant compared to other uncertainties associated with your approach, though.

REFERENCE:

*Räisänen, P., Kokhanovsky, A., Guyot, G., Jourdan, O., and Nousiainen, T.: Param-*

*eterization of single-scattering properties of snow, The Cryosphere, 9, 1277–1301, https://doi.org/10.5194/tc-9-1277-2015, 2015.*

9. p. 10, lines 20-21: "The wavelength range of the estimation an the $\eta_{\mathrm{mes}}$ do not impact the eqBC retrieval". Do you mean "...do not impact significantly"?

10. p. 11, line 5: The range of dust MAE (0.071–0.127 $\mathrm{m^2\,g^{-1}}$ at 407 nm) seems a bit conservative, considering that you mention an order of magnitude uncertainty on p. 2, line 25, and that the whole range of Table 4 in Caponi et al. (2017) goes from 0.071 to 0.621 $\mathrm{m^2\,g^{-1}}$ (even though the maximum represents Sahel only).

11. p. 11, line 28: "The variations of $g$ do not impact LAP retrievals". Again, "do not impact significantly"? At any rate, this seems surprising to me, especially when you first explain that while the ratio $B/(1-g)$ may be fixed at 10.7, variations of $B$ and $g$ could still have an impact as $B(1-g)$ may vary. What was the actual range of $g$ and $B$ considered when you arrived at this conclusion?

12. p. 11, line 29: Please specify that you mean the imaginary part of the ice refractive index ($n_i$).

13. p 12, section 4.4: Also mention that according to Fig. 12, the estimated dust fraction to LAP absorption is underestimated in almost all cases (this might tell something about errors in the spectral signature of BC vs. dust absorption, even if pursuing this issue further is not feasible here).

14. p. 15, line 6: I think AART should be mentioned already in the theory section 3.4.

15. p. 15, line 30: "Using Monte Carlo ray tracing on real micro-tomography snow samples." I think it would be appropriate to mention here explicitly the concept of close-packing. In fact, a recent paper by He et al. (2017) suggests that close-packing of snow

may substantially enhance the albedo reduction caused by BC in snow (and hence the total absorption in snow). However, my intepretation of their paper is that this mainly happens because close-packing makes the effective snow grain size larger, or the SSA smaller, so that radiation penetrates deeper into snow (which is an effect that should be captured even by traditional 1D radiative transfer). What do you think?

REFERENCE:

*He, C., Takano, Y., and Liou, K.-N. (2017), Close packing effects on clean and dirty snow albedo and associated climatic implications, Geophys. Res. Lett., 44, 3719–3727, doi:10.1002/2017GL072916.*

16. Caption of Fig. 1: "B, g, LAP MAE" is quite cryptic because these parameters appear in the text much later than Fig. 1 is introduced. If you replaced this with "$B = 1.6$, $g = 0.85$, LAP MAEs defined in Sect. 3.4.2" it would already be much more explicit.

17. Fig. 2: I am puzzled about the numerical values here. Ice absorption coefficient reaches down to $10^{-6}$m$^{-1}$ at 390 nm. In Picard et al. 2016 (The Cryosphere, 10, p. 2655–2672), the lowest values for the IA2008 curve (which is probably too low) are slightly below $10^{-3}$ m$^{-1}$, i.e., three orders of magnitude higher. Also, what is assumed about snow density here?

18. In Figs. 3 and 7, it would be logical to switch the colors for 550 and 700 nm (as the wavelength for red light is ≈700 nm, and green light ≈550 nm).

19. In Fig. 8 and 10, can you include a scale showing how the size is related to the maximum wavelength of the AFEC estimation?

[Figure]

**Technical and language corrections**

1. p. 1, line 14: replace "dependence" with "sensitivity".

2. p. 9, line 17: this should be "dust source regions".

3. p. 9, line 19: replace "inferior to" with "smaller than".

4. The order of figures differs from the order they are cited in the text. Fig. 5 is cited first time after Figs. 6 and 7, and Fig. 14 is cited first time before Figs. 12 and 13.

5. p. 12, line 13: replace "few number" with "small number".

6. p. 13, line 27: this should be "abnormally"

7. In Figs. 2, 3b, 4, 6: There is a label missing on the lower left corner, and should be added so that the reader can interpret the scale accurately. (Hint: this is probably a round-off problem with your graphics software. But graphics software can be cheated: e.g., in Fig. 2, try to start the scale from $9.99 \times 10^{-7}$ instead of $10^{-6}$!).

8. Fig. 3: Add units of depth (m) on the $y$-axis.

9. In the caption of Fig. 10, "comporting" sounds like a strange choice of verb.

10. In Fig. 12, $x$-axis label, "Mesured" should be "Measured".

---

## Referee Comment (RC2) · Anonymous Referee #2 · 17 Jun 2019

This study describes a novel and rapid technique to make in-situ measurements of the vertical profile of light absorbing impurities in snow. The technique relies on spectral irradiance measurements conducted via a narrow probe that is slowly inserted into the snow. Because the technique relies purely on radiative transfer theory, it does not require snow samples to be transported to the laboratory for chemical measurements. The underlying theory is nicely presented, and although the technique 'should' work well in principle, as with many ideal techniques there is substantial bias between the theoretically-derived and directly-measured impurity contents, as clearly acknowledged by the authors. The study presents a nice exploration of sources of uncertainty via parameter perturbations, and as far as I can tell the study has adequately explored all likely sources of bias. Unsurprisingly, the optical properties of BC and dust, which

must be known apriori for this technique, are plausible culprits for the bias. Real uncertainty and variability in these properties could, by themselves, explain much of the reported bias. Overall, this is a very thorough and well-written paper describing a novel technique, and I recommend publication after the minor issues described below are addressed.

General issues:

Equation 1: It is noted that Phi represents the dust -> eqBC conversion function but this function is not really described in much detail. Please elaborate on what precisely this function is and/or how it is calculated. A related question is: Why is the eqBC vs dust line shown in Figure 1 not perfectly linear? This suggests that the conversion function is not so simple.

Minor issues:

p3, lines 26-28: "Picard et al (2016) ... meaning that SIP measurements could be an order of magnitude more sensitive to LAP than albedo measurements." - This statement implies that BC concentrations less than 50 ng/g cannot be detected via albedo measurements. This threshold seems a bit high, especially for visible wavelengths. Are you referring to broadband albedo? Please clarify or justify.

p6, line 10: "It is to note" -> "It is noteworthy"

p6, line 12: "... the unit of ng/g eqBC refers to 1 ng/g of eqBC concentration" - This seems either unnecessarily obvious or needs elaboration.

p7, line 8: "ice matrix surface (m2)" -> "ice matrix surface area (m2)"

p7, Eqns 10 and 11: It is a bit confusing that sigma_a and gamma both represent absorption coefficients of ice. It appears that sigma_a is the absorption coefficient of "snow due to ice", whereas sigma is the absorption coefficient of bulk ice. Please clarify the wording to communicate this.

[Figure]

p7, Eqn 10: Maybe clarify that rho is the density of snow, if this has not already been done.

p10, line 3: "did not fit well the" -> "did not fit well with the"

p10, line 27: Please clearly communicate the sign of the bias. i.e., Was the chemically-determined or SOLEXS-derived BC estimate higher?

p12, line 6: "an higher" -> "a higher"

p13, line 2: "the radiative impact" -> "the calculated radiative impact", correct? Or if not, please clarify this sentence, again with respect to the sign of the bias (higher derived-BC or chemically-measured BC?).

p13: line 27: "In some case, an abnormally" -> "In some cases, an abnormally"

p14, line 15: "clearly break" -> "clearly breaks" or better "clearly violates"

p14, line 24: "more impacting" -> "more impact"

Figures 6, 13 and 14: In the legend, why does one curve show BC and the other rBC? Please remind readers of why this distinction is needed here. It seems confusing and potentially unnecessary.

---

## Author Comment (AC1) · 5 Jul 2019

First of all, we slightly modified the results in the new version of the manuscript. The retrieval method used in the last version indeed included a small regularization term to minimise the SSA difference between the retrieval and the measured value contrary to what is written from page 10 lines 18 to 21. As this term had a small weight the impacts on our results are small. However to remain consistent with the description of the method, the regularization term has been removed and all figures and paragraphs impacted have been modified.

The impacts on our results are a small improvement of LAP retrieval performances ($r^2$ 0.74→0.80, mainly due to one point shifting under the sensitivity threshold of 5ng/g) and a small reduction of SSA retrieval performances ($r^2$ 0.73→0.71). The conclusion of the paper remains unchanged.

Answer to Anonymous Referee #1 (Referee):

We would like to thank Anonymous Referee #1 for his extensive analysis of our manuscript which helps improving our paper. All the comments have been addressed and point by point response is provided below each comment.
The reviewer initial comments are written in black, our answer in blue and the corrections in the paper are highlighted in red. The line numbers which are used in the answers correspond to the new version of the manuscript.

**General comments:**

This paper presents an original technique for estimating the concentration of light absorbing particles (LAP) in snow based on measurements of vertical profiles of spectral irradiance made at wavelengths from 350 to 950 nm.

The approach is based on values of the asymptotic flux extinction coefficient (AFEC) derived for homogeneous layers in snow. An optimization algorithm is developed that estimates the snow specific surface area (SSA) and concentrations of dust ($c_{dust}$) and black carbon ($c_{BC}$) based on the AFEC spectra. The inferred parameters are then compared with independent observations of SSA and equivalent-BC. It is shown that the estimated equivalent-BC concentration correlates quite well with that inferred from chemical measurements, although only for concentrations larger than 5 ng g $-1$ . However, disturbingly, there is a substantial systematic bias of ≈16 ng g $-1$ between the equivalent-BC concentrations inferred from the algorithm and the chemical measurements. Various sources of error are discussed, but they are able to explain the bias only partially.

The proposed method would allow for a relatively fast measurement of the LAP concentrations (compared to chemical analyses of snow samples), but at this stage, its attractiveness is reduced by the presence of a systematic bias that is not properly understood. Also, the applicability of the method is limited to homogeneous layers, and it can only be used at depths where the radiation field in snow is diffuse, and sufficiently far away from the underlying ground. Therefore, rather than providing a method of measuring LAPs that is ready to use at its present state, I view the research reported in this article as a "closure experiment" that probes our understanding of both the radiative transfer in snow with LAPs and of the measurements. Clearly, this understanding is still less than perfect.

Even though the proposed method does not yet work fully satisfactorily, I think this is an innovative and interesting study. I therefore recommend its publication in the Cryosphere, subject to the minor comments listed below.

**Specific comments :**

1. p. 2, line 2: Can you specify what you mean with "near infrared"? Different definitions exist. Wikipedia mentions both 1.4 μm and 2.5 μm, while in atmospheric radiative transfer literature, NIR usually extends to 4 μm. It is well known that snow albedo is generally quite low at wavelengths larger than ≈1.4 μm.

Indeed, the definition of near infrared can be ambiguous. Here by near infrared we mean the wavelength range from the visible to ≈1.4 μm. The first sentence of the introduction has been modified in the revised manuscript:

Snow is a highly reflective medium in the wavelengths of the visible and of the near infrared (up to 1.4 μm, referred to as NIR in the following) where most of the solar energy is available (Warren, 1982).

2. p. 4, line 11: do you mean the onset of the snow season, or onset of the snow melt season?

Here it should be understood: "from the onset of the snow season". It has been explicitly added in the manuscript p4 l.11:

both in winter and spring conditions from the onset of the snow season to the total melt-out of the snowpack.

3. p. 4, lines 17–18: "taking precautions not to enlarge the hole". Even so, some light could leak deeper into snow through the small space of air between the rod and the snowpack. Did you eliminate this somehow?

Following the protocol described in Picard et al. 2016, we systematically added a few millimeters of snow on the surface around the rod to cover the void space from direct sun beam. This avoid the risk of direct light penetration in the small space of air between the rod and the snowpack though we cannot fully exclude that some additional radiation is scattered in the hole. We modified the manuscript accordingly :

page 4 line 18 :'taking precautions not to enlarge the hole. A few millimeters of snow was systematically added on the surface around the rod to shield the void space from direct sun beam in order to minimise the leak of additional solar radiation into the hole.'

4. p. 6, line 11: It would be helpful to show the spectral Mass Absorption Efficiencies (MAE) of BC and dust assumed in the computation of equivalent BC concentration in the conventional units (m2 g−1). I can see that related information is given in Fig. 2, but it requires effort to do the conversion.

Indeed, this information is of importance and is not illustrated in the manuscript. The Figure 1 and his caption has been modified as follow to account for this remark:

[Figure]

Figure 1. a) Mass Absorption Efficiency (MAE) values of BC and dust used in the present study as a function of wavelength. b) EqBC concentration corresponding to a given dust concentration using these MAE values and the methods described in section 3.1.

Consequently, all references to Figure 1 have been replaced by Figure 1 b) in the manuscript and the text p.6 l.13 has also been modified as follows:
It is noteworthy that the function $\Psi$ has a strong dependence to the spectral distribution of the incident solar radiation and on the radiative transfer model parameters, mainly on the selected values of BC and dust Mass Absorption Efficiency (MAE). These MAE values are represented in Figure 1 a) and detailed in section 3.4.2.

5. p. 6, Eq. (3): please define z and z0. Presumably z is the depth, increasing downwards (even though in Figs. 3 and 7, the values are negative).

Indeed z is the depth increasing downwards and $z_0$ is a reference depth.
The text p.6 l.28 has been modified as follows:
Following the radiative transfer theory in a homogeneous layer far from any interface, the intensity at a given wavelength λ, I(z,λ), decreases exponentially with depth. This writes:
$I(z,\lambda) = I(z0,\lambda)e^{-ke(\lambda)(z-z0)}$,
where ke(λ) is the Asymptotic Flux Extinction Coefficient (AFEC) expressed in $m^{-1}$, z is the depth increasing downwards and z0 is a reference depth.

Moreover, the values on y axis of Fig 3 and 6 have been noted positive to ensure consistency throughout the manuscript.

6. p. 6, line 27: Did you apply some quantitative criteria for the minimum distance between your zones-of-interest and ground?

As explained in the manuscript, shallow snowpacks where ground is expected to have a significant contribution to the total energy absorption are discarded. For this we use the quantitative criterion that total snowdepth must be higher than 50cm (already indicated in the manuscript page 7 line 4). Nevertheless even for snowpack thicker than 50 cm, the near proximity of the ground (few centimeters) might impact the radiation field. In the present manuscript we did not established any quantitative criterion to discard these cases because the minimal ground-ZOI distance is of 18 centimeters which we consider too far to impact significantly our result. This has been explicitely added in the manuscript page 7 line 4

"Note that the minimum distance between the ZOI and the ground is of 18 cm, which we believe thick enough to prevent any significant disturbance of the measured signal due to the presence of the ground."

7. p. 8, Eq. (6): please define B (absorption enhancement parameter?).

Indeed B is the absorption enhancement parameter. It has been added in the manuscript p.8 l.19:

with $\sigma_a(m^{-1})$ the absorption coefficient of snow due to ice and B the absorption enhancement parameter.

8. p. 9, Eq. (12): Strictly speaking, g and B also depend on wavelength. An example of this can be seen in Figure 6a,c of Räisänen et al. (2015) for a few assumptions about snow grain shapes. (I think their parameter is equal to B in this study). For the wavelengths of interest for the present study (350-950 nm), B(1 – g) might vary up to ±5% as a function of wavelength (with largest values at the short wavelengths). I guess this would be insignificant compared to other uncertainties associated with your approach, though.

The spectral variations of B and g were neglected in this study since we believed the impact was negligible. However the assumption was not clearly formulated in the manuscript and your remark pointed out the relevance to assess this impact on our method. As the implementation of B and g spectral variations in our algorithm was quite straightforward, we implemented B and g spectral variations as in Appendix F of Libois 2014 PhD, which are taken from Kokhanovsky (2004). As expected the impact on our retrieval is minor, which is illustrated on the two figures hereafter.

[Figure]

Accounting for B and g spectral variation (new version)          B ang g constant over wavelengths (old version)

We decided to keep the spectral variations of B and g in the manuscript. It is now explained in the manuscript p.10 l.5.

The snow shape parameters B and g are constant over time and for all types of snow. These parameters have a small dependence to the wavelength λ implemented following Kokhanovsky (2004) and Appendix F of Libois (2014). This dependence is a function of the real part of ice refractive index ri which is taken from Warren and Brandt 2008 and is written as follows:
$B(\lambda)=B_0+0.4\ (r_i(\lambda)-1.3)$
$g(\lambda)=g_0-0.38\ (r_i(\lambda)-1.3)$
The absorption enhancement parameter $B_0$ is set to 1.6 and the asymmetry factor $g_0$ is set to 0.85, considered to be good approximations to describe all type of snow (Libois et al 2014b). As the spectral dependence of $B(\lambda)$ and $g(\lambda)$ is small over the range of wavelength targeted by this study, they are referred as B and g for sake of simplicity.

9. p. 10, lines 20-21: "The wavelength range of the estimation an the ηmes do not impact the eqBC retrieval". Do you mean "...do not impact significantly"?

By this sentence we meant that neither the wavelength range used for the retrieval estimation nor the value of ηmes are found to be correlated with the accuracy of the retrieval.. Given the uncertainties associated to our method it is not possible to guarantee that there is absolutely no impact of these two parameters on the retrieval accuracy. Consequently the sentence has been modified in the manuscript p.11 l.9:

Neither the wavelength range used for the estimation nor the value of ηmes are found to be correlated with the accuracy of the retrieval.

10. p. 11, line 5: The range of dust MAE (0.071–0.127 m2 g−1 at 407 nm) seems a bit conservative, considering that you mention an order of magnitude uncertainty on p. 2, line 25, and that the whole range of Table 4 in Caponi et al. (2017) goes from 0.071 to 0.621 m2 g−1 (even though the maximum represents Sahel only).

Indeed the range of dust MAE is quite small compared to the maximum that can be found in the literature. However as you stated, these maximum are found for dust coming from regions that less likely to affect our study area.
A paragraph clarifying this point has been added in the results p.11 l.24:

Caponi et al (2017) suggest that for dust particles smaller than 2.5 µm (PM2.5), which is the major dust type in regard of measured size distribution, dust MAE at 407 nm is between 0.071 and 0.127 $m^2\ g^{-1}$ (0.103 for Figure 8) for north Saharan dust. It should be noted that higher values of dust MAE can be found in the literature and in turn higher uncertainties associated to this parameter could be considered. However, these values corresponds to source regions that less likely affect our study area (e.g. up to 0.6 $m^2\ g^{-1}$ for Sahel desert, Caponi et al. 2017).

11. p. 11, line 28: "The variations of g do not impact LAP retrievals". Again, "do not impact significantly"? At any rate, this seems surprising to me, especially when you first explain that while the ratio B/(1 − g) may be fixed at 10.7, variations of B and g could still have an impact as B(1−g) may vary. What was the actual range of g and B considered when you arrived at this conclusion?

Under the hypothesis made in our study and following our method, this conclusion is valid for any range of B and g. Indeed, when looking at Equation 12

$$k_e(\lambda) \approx \sqrt{\frac{3(1-g)}{2}\rho^2 SSA\left(\frac{B\gamma_{ice}(\lambda)}{\rho_{ice}} + \sum_i MAE_i(\lambda)c_i\right)}.$$

the impurity retrieval is totally independent of the value of g as SSA is let as a free parameter of our optimization scheme. Any change of can be fully compensated by a change in our SSA retrieval. In turn g variation have only an impact on our SSA retrieval and not on LAP retrieval. This has been clarified in the new version of the manuscript ( page 12 line 21) :
"The variations of g do not impact LAP retrievals since SSA is left as a free parameter in our method and can counterbalance any variation of g (see Eq. 12). "

12. p. 11, line 29: Please specify that you mean the imaginary part of the ice refractive index (ni).

The correction has been added to the manuscript p.9 l.18 and p.12 l.23.:
"the refractive index of ice" has been replaced by "the imaginary part of the refractive index of ice"

13. p 12, section 4.4: Also mention that according to Fig. 12, the estimated dust fraction to LAP absorption is underestimated in almost all cases (this might tell something about errors in the spectral signature of BC vs. dust absorption, even if pursuing this issue further is not feasible here).

That is an interesting point and it is now mentioned in the manuscript p.13 l.8.
The estimated dust fraction is almost systematically lower than the measured dust fraction (12/14 points). This may either indicate that the relative absorption of dust versus BC used in this study could be improved or that there are systematic biases in dust or rBC measurements.

14. p. 15, line 6: I think AART should be mentioned already in the theory section 3.4.

The mistake has been corrected and a sentence has been modified in the Section 3.4 to explicitly mention AART. p.8 l.11
The Asymptotic Approximation of the Radiative Transfer theory (AART; Kokhanovsky and Zege, 2004)) for pure snow shows that for convex crystals :

15. p. 15, line 30: "Using Monte Carlo ray tracing on real micro-tomography snow samples." I think it would be appropriate to mention here explicitly the concept of closepacking. In fact, a recent paper by He et al. (2017) suggests that close-packing of snow may substantially enhance the albedo reduction caused by BC in snow (and hence the total absorption in snow). However, my intepretation of their paper is that this mainly happens because close-packing makes the effective snow grain size larger, or the SSA smaller, so that radiation penetrates deeper into snow (which is an effect that should be captured even by traditional 1D radiative transfer). What do you think?

The effect of the close packing as found by C. He et al. (2017) and the consequence on the extinction in snow under the presence of impurities are in line with our finding of the overestimation by the optical method. However, we have concerns about the findings in C. He et al. (2017) which do not discuss about a large corpus of work in the 90s by Mischenko et al., showing that in snow (very large particles compared to the wavelength) the close packing effect is negligible. These opposite conclusions and the absence of citation of this previous work make us conclude that this subject is still debated and needs to be confirmed. This is the reason why we prefer not to address this aspect in the present manuscript.

16. Caption of Fig. 1: "B, g, LAP MAE" is quite cryptic because these parameters appear in the text much later than Fig. 1 is introduced. If you replaced this with "B = 1.6, g = 0.85, LAP MAEs defined in Sect. 3.4.2" it would already be much more explicit.

The caption of Figure 1 b) has been modified to account for this remark:
b) EqBC concentration corresponding to a given dust concentration using these MAE values and the methods described in section 3.1

17. Fig. 2: I am puzzled about the numerical values here. Ice absorption coefficient reaches down to 10−6m−1 at 390 nm. In Picard et al. 2016 (The Cryosphere, 10, p. 2655–2672), the lowest values for the IA2008 curve (which is probably too low) are slightly below 10−3 m−1, i.e., three orders of magnitude higher. Also, what is assumed about snow density here?

Figure 2 is supposed to represent $\sigma_a$ for snow and $\sigma_a$ for LAP. However, the legend and the caption implied that ice absorption ($\gamma_{ice}$) was represented instead of snow absorption due to ice. Moreover the values actually represented were $\sigma_a/\rho$. This choice was made to be independent of the snow density value but the explanation was not in the manuscript. As this choice is inconsistent with the writing of the equations, the value represented are now $\sigma_a$ and not $\sigma_a/\rho$. To this end, a density hypothesis of 200 kg m$^{-3}$ for snow has been done and is specified in the caption. Moreover, the legend and the caption of the figure were modified to replace "ice absorption" by "snow absorption".
Finally, in Picard et al. 2016 (The Cryosphere, 10, p. 2655–2672), the values represented are the one of pure ice absorption ($\gamma_{ice}$) and not the one of snow absorption as in the present study, explaining the differences.

The caption and legend of Figure 2 have been modified as follows:

[Figure]

Figure 2. Spectral signature of the absorption coefficients $\sigma_a$ for snow and different types of LAPs assuming a snow density of 200 kg m$^{-3}$.

18. In Figs. 3 and 7, it would be logical to switch the colors for 550 and 700 nm (as the wavelength for red light is 700 nm, and green light 550 nm).

Thank you for this suggestion, the modification has been done. A different marker has also been used for each wavelength to facilitate black and white reading. Moreover, according to your comment 5 the y axis have been modified to have positive depth.

19. In Fig. 8 and 10, can you include a scale showing how the size is related to the maximum wavelength of the AFEC estimation?

This modification has been added to Figure 8 and 10 as follows:

[Figure]

**Technical and language corrections :**

All the remarks of this section have been corrected in the manuscript; additional information can be found after the comment when necessary.

1. p. 1, line 14: replace "dependence" with "sensitivity".

2. p. 9, line 17: this should be "dust source regions".

3. p. 9, line 19: replace "inferior to" with "smaller than".

4. The order of figures differs from the order they are cited in the text. Fig. 5 is cited first time after Figs. 6 and 7, and Fig. 14 is cited first time before Figs. 12 and 13.

5. p. 12, line 13: replace "few number" with "small number".

6. p. 13, line 27: this should be "abnormally"

7. In Figs. 2, 3b, 4, 6: There is a label missing on the lower left corner, and should be added so that the reader can interpret the scale accurately. (Hint: this is probably a round-off problem with your graphics software. But graphics software can be cheated: e.g., in Fig. 2, try to start the scale from $9.99 \times 10^{-7}$ instead of $10^{-6}$!).

8. Fig. 3: Add units of depth (m) on the y-axis.

9. In the caption of Fig. 10, "comporting" sounds like a strange choice of verb.

The verb comporting has been replace by "with concomitant measurement" in caption of Figure 10 and 11 to be consistent with Figures 8 and 9.

*10. In Fig. 12, x-axis label, "Mesured" should be "Measured".*

**References:**

Dong, Z., Kang, S., Qin, D., Shao, Y., Ulbrich, S., & Qin, X. (2018). Variability in individual particle structure and mixing states between the glacier–snowpack and atmosphere in the northeastern Tibetan Plateau. The Cryosphere, 12(12), 3877-3890.

He, C., Li, Q., Liou, K. N., Takano, Y., Gu, Y., Qi, L., ... & Leung, L. R. (2014). Black carbon radiative forcing over the Tibetan Plateau. Geophysical Research Letters, 41(22), 7806-7813.

He, C., Takano, Y., and Liou, K.-N. (2017), Close packing effects on clean and dirty snow albedo and associated climatic implications, Geophys. Res. Lett., 44, 3719–3727, doi:10.1002/2017GL072916.

Kokhanovsky, A. A. (2004). Light scattering media optics. Springer Science & Business Media.

Libois Q. Evolution des propriétés physiques de neige de surface sur le plateau Antarctique. Observations et modélisation du transfert radiatif et du métamorphisme. 2014. Thèse de doctorat. Grenoble.

Libois, Q., Picard, G., Dumont, M., Arnaud, L., Sergent, C., Pougatch, E., Sudul, M., and Vial, D.: Experimental determination of the absorption enhancement parameter of snow, Journal of Glaciology, 60, 714–724, 2014.

Räisänen, P., Kokhanovsky, A., Guyot, G., Jourdan, O., and Nousiainen, T.: Parameterization of single scattering properties of snow, The Cryosphere, 9, 1277–1301, https://doi.org/10.5194/tc-9-1277-2015, 2015.

Warren, S. G., & Brandt, R. E. (2008). Optical constants of ice from the ultraviolet to the microwave: A revised compilation. Journal of Geophysical Research: Atmospheres, 113(D14).

---

## Author Comment (AC2) · 5 Jul 2019

First of all, we slightly modified the results in the new version of the manuscript. The retrieval method used in the last version indeed included a small regularization term to minimize the SSA difference between the retrieval and the measured value contrary to what is written from page 10 lines 18 to 21. As this term had a small weight the impacts on our results are small. However to remain consistent with the description of the method, the regularization term has been removed and all figures and paragraphs impacted have been modified.

The impacts on our results are a small improvement of LAP retrieval performances ($r^2$ 0.74→0.80, mainly due to one point shifting under the sensitivity threshold of 5ng/g) and a small reduction of SSA retrieval performances ($r^2$ 0.73→0.71). The conclusion of the paper remains unchanged.

Answer to Anonymous Referee #2 (Referee):

We would like to thank Anonymous Referee #2 for his pertinent comments pointing out some issues in our manuscript. The comments have been addressed and discussed hereafter.
The reviewer initial comments are written in black, our answer in blue and the corrections in the paper are highlighted in red. The line numbers which are used in the answers correspond to the new version of the manuscript.

This study describes a novel and rapid technique to make in-situ measurements of the vertical profile of light absorbing impurities in snow. The technique relies on spectral irradiance measurements conducted via a narrow probe that is slowly inserted into the snow. Because the technique relies purely on radiative transfer theory, it does not require snow samples to be transported to the laboratory for chemical measurements. The underlying theory is nicely presented, and although the technique 'should' work well in principle, as with many ideal techniques there is substantial bias between the theoretically-derived and directly-measured impurity contents, as clearly acknowledged by the authors. The study presents a nice exploration of sources of uncertainty via parameter perturbations, and as far as I can tell the study has adequately explored all likely sources of bias. Unsurprisingly, the optical properties of BC and dust, which must be known apriori for this technique, are plausible culprits for the bias. Real uncertainty and variability in these properties could, by themselves, explain much of the reported bias. Overall, this is a very thorough and well-written paper describing a novel technique, and I recommend publication after the minor issues described below are addressed.

**General issues:**

Equation 1: It is noted that Phi represents the dust -> eqBC conversion function but this function is not really described in much detail. Please elaborate on what precisely this function is and/or how it is calculated. A related question is: Why is the eqBC vs dust line shown in Figure 1 not perfectly linear? This suggests that the conversion function is not so simple.

$\Psi$ indeed represents the dust -> eqBC conversion function. This function is computed as follows:

1) The energy absorbed by a semi-infinite snowpack containing a given quantity of dust between 350 and 900 nm is computed for dust concentration spanning the whole range of observations (0→25$\mu$ g g$^{-1}$). The incoming spectral repartition of the energy is done with SBDART as explained in the text.
2) An optimization is ran to find which BC quantity would lead to the same broadband energy absorption between 350 and 900 nm with a resolution of 1 nm.

The equivalent BC concentration computed at a single wavelength, $\lambda$, by our method would be linear. Indeed following Equation 11, the equivalent BC concentration can be expressed as:
$C_{eqBC}=(c_{dust} MAE_{dust}(\lambda )) / MAE_{BC}(\lambda )$
The non-linearity is introduced by the spectral dependence of the ratio between $MAE_{dust}(\lambda )$ and $MAE_{BC}(\lambda )$. As the estimated eqBC concentration is the integrated absorption on several wavelengths and the absorption has a non-linear response to LAP concentration , $\Psi$ has no reason to be linear. Moreover, strictly speaking, $\Psi$ also depends on BC concentration, on the SSA of the snowpack and on the selected spectral solar irradiance.
 This has been clarified in the manuscript p.6 l.10 .

To do so, the energy absorbed by a semi-infinite snowpack with a SSA of 15 m$^2$ kg$^{-1}$ is computed at each wavelength between 350 and 900nm. The spectral incoming irradiance is computed with the detailed atmospheric radiative model SBDART (Ricchiazzi et al., 1998), for mid-latitude winter in clear sky conditions .
It is noteworthy that the function $\Psi$ has a strong dependence to the spectral distribution of the incident solar radiation and on the radiative transfer model parameters, mainly on the selected values of BC and dust Mass Absorption Efficiency (MAE). These MAE values are represented in Figure 1 a) and detailed in section 3.4.2. Strictly

speaking, Ψ also depends on the BC concentration and on the SSA of the snowpack but this minor impact is neglected here.

**Minor issues:**

p3, lines 26-28: "Picard et al (2016) ... meaning that SIP measurements could be an order of magnitude more sensitive to LAP than albedo measurements." - This statement implies that BC concentrations less than 50 ng/g cannot be detected via albedo measurements. This threshold seems a bit high, especially for visible wavelengths.
Are you referring to broadband albedo? Please clarify or justify.

Indeed this threshold is bit overestimated. It was based on Zege et al.2011 and Warren et al. 2013 but their paper applies only to remote sensing retrieval and terrestrial spectral albedo measurements can be expected to have a better accuracy and then sensitivity.
The sentence has been modified in the manuscript p.3 l.27

"..meaning that SIP measurements could be more sensitive to LAP than albedo measurements."

p6, line 10: "It is to note" -> "It is noteworthy"

The correction has been accounted for.

p6, line 12: "... the unit of ng/g eqBC refers to 1 ng/g of eqBC concentration" – This seems either unnecessarily obvious or needs elaboration.

This sentence (p.6 l.17) has been replaced by:
In the following, the LAP concentrations are expressed in ng $g^{-1}$ eqBC.

p7, line 8: "ice matrix surface (m2)" -> "ice matrix surface area (m2)"

The correction has been accounted for.

p7, Eqns 10 and 11: It is a bit confusing that sigma_a and gamma both represent absorption coefficients of ice. It appears that sigma_a is the absorption coefficient of "snow due to ice", whereas gamma is the absorption coefficient of bulk ice. Please clarify the wording to communicate this.

Indeed the explanation on this two variables was confusing. It has been clarified as follows when the variables are introduced: P.8 L.19
with $\sigma_a$ ($m^{-1}$) the absorption coefficient of snow due to ice and B the absorption enhancement parameter. The term $\gamma(\lambda)$ ($m^{-1}$) is the absorption coefficient of bulk ice and is related to the imaginary part of ice refractive index $n_i(\lambda)$ as follows:

p7, Eqn 10: Maybe clarify that rho is the density of snow, if this has not already been done.

The variable ρ was not introduced in our equations. It has been corrected after Equation 5 (P.8 l.16)
where ρ is the density of snow and SSA is its Specific Surface Area ($m^2$ $kg^{-1}$ Legagneux et al. 2002)

p10, line 3: "did not fit well the" -> "did not fit well with the"

The correction has been accounted for.

p10, line 27: Please clearly communicate the sign of the bias. i.e., Was the chemically-determined or SOLEXS-derived BC estimate higher?

The sentence mentioning the bias has been completed as follows p.11 l.16
Indeed, the correlation in this range has a $r^2$ of 0.81 in spite of a significant bias of 14.6 ng $g^{-1}$ eqBC; the chemically measured concentrations being lower than the SOLEXS retrieval

p12, line 6: "an higher" -> "a higher"

The correction has been accounted for.

p13, line 2: "the radiative impact" -> "the calculated radiative impact", correct? Or if not, please clarify this sentence, again with respect to the sign of the bias (higher derived-BC or chemically-measured BC?).

p13: line 27: "In some case, an abnormally" -> "In some cases, an abnormally"

The correction has been accounted for.

p14, line 15: "clearly break" -> "clearly breaks" or better "clearly violates"

The correction has been accounted for, "clearly break" has been replaced by "clearly violates".

p14, line 24: "more impacting" -> "more impact"

The correction has been accounted for.

Figures 6, 13 and 14: In the legend, why does one curve show BC and the other rBC? Please remind readers of why this distinction is needed here. It seems confusing and potentially unnecessary.

In these figures, the distinction is done between BC and rBC because rBC is the quantity measured by the SP2 which is just one way among others to measure BC in snow. As these different measurement techniques can strongly diverge in term of BC concentration (e.g., Lim et al. 2014) this information is important. The following sentence has been added in the caption of Figure 5 of the manuscript.

Note that rBC is the refractory BC concentration measured by $SP^2$ instrument

It is noteworthy that Figure 5 of the new manuscript was the Figure 6 you mentioned in your comment.

References:

Zege, E. P., Katsev, I. L., Malinka, A. V., Prikhach, A. S., Heygster, G., & Wiebe, H. (2011). Algorithm for retrieval of the effective snow grain size and pollution amount from satellite measurements. *Remote Sensing of Environment*, *115*(10), 2674-2685.

Warren, S. G. (2013). Can black carbon in snow be detected by remote sensing?. *Journal of Geophysical Research: Atmospheres*, *118*(2), 779-786.

Lim, S., Faïn, X., Zanatta, M., Cozic, J., Jaffrezo, J. L., Ginot, P., & Laj, P. (2014). Refractory black carbon mass concentrations in snow and ice: method evaluation and inter-comparison with elemental carbon measurement. *Atmospheric Measurement Techniques*, *7*(10), 3307-3324.